# Structure of the CLC-1 chloride channel from *Homo sapiens*

**Eunyong Park, Roderick MacKinnon\***

Laboratory of Molecular Neurobiology and Biophysics, Howard Hughes Medical Institute, The Rockefeller University, New York, United States

**Abstract** CLC channels mediate passive $Cl^-$ conduction, while CLC transporters mediate active $Cl^-$ transport coupled to $H^+$ transport in the opposite direction. The distinction between CLC-0/1/2 channels and CLC transporters seems undetectable by amino acid sequence. To understand why they are different functionally we determined the structure of the human CLC-1 channel. Its 'glutamate gate' residue, known to mediate proton transfer in CLC transporters, adopts a location in the structure that appears to preclude it from its transport function. Furthermore, smaller side chains produce a wider pore near the intracellular surface, potentially reducing a kinetic barrier for $Cl^-$ conduction. When the corresponding residues are mutated in a transporter, it is converted to a channel. Finally, $Cl^-$ at key sites in the pore appear to interact with reduced affinity compared to transporters. Thus, subtle differences in glutamate gate conformation, internal pore diameter and $Cl^-$ affinity distinguish CLC channels and transporters.

DOI: https://doi.org/10.7554/eLife.36629.001

## Introduction

Transporters – also known as pumps – and channels both mediate the transfer of ions and molecules across biological membranes. But the two are thermodynamically contrasting: transporters require the input of external energy while channels are passive, meaning the substrate simply diffuses down its electrochemical gradient. Except in rare cases, transporters and channels correspond to separate, unrelated structural families. CLC proteins are one of the exceptions. Channel-forming CLCs are passive $Cl^-$ conductors (*Jentsch et al., 1990*; *Miller and White, 1984*), while transporter-forming CLCs exchange, with fixed stoichiometry, two $Cl^-$ ions and one proton ($H^+$) in opposite directions (i.e., they are $Cl^-/H^+$ antiporters) (*Accardi and Miller, 2004*; *Picollo and Pusch, 2005*; *Scheel et al., 2005*). The external energy input in CLC transporters comes from the energetic coupling of the transported ions, $Cl^-$ and $H^+$, such that the electrochemical gradient of one ion drives movement of the other. The puzzling aspect of this dual functionality within the CLC protein family is that at the level of amino acid sequence, the distinction between the channels and transporters is not apparent.

Conceptually, the distinction between channels and transporters in general has been explained in terms of gating models that invoke one or two primary gates: channels are described as pores with one gate and transporters as pores with two gates that are never permitted to open simultaneously (*Figure 1*) (for review see [*Gadsby, 2009*]). While it is true that channels and transporters are most often unrelated structurally, the gating model description implies that, in principle, similar structures could give rise to both, as one can imagine that a transporter could become a channel if one or both gates are compromised. CLC channels seem to fall under this category of channels that emerged from a family of transporters (*Accardi and Picollo, 2010*; *Lísal and Maduke, 2008*; *Miller, 2006*).

Structural and functional studies support a plausible mechanistic model for the operation of CLC transporters. CLC transporter structures show a narrow $Cl^-$ transport pathway with three consecutive $Cl^-$-binding sites, referred to as $S_{ext}$, $S_{cen}$ and $S_{int}$, for external (nearest the extracellular solution), central and internal (nearest the intracellular solution), respectively. Chloride is observed at

**\*For correspondence:**
mackinn@mail.rockefeller.edu

**Competing interests:** The authors declare that no competing interests exist.

**eLife digest** Channels and transporters are two classes of proteins that transport molecules and ions – collectively referred to as "substrates" – across cell membranes. Channels form a pore in the membrane and the substrates diffuse through passively. Transporters, on the other hand, actively pump substrates across a membrane, consuming energy in the process. Thus, channels and transporters work in distinct ways.

Channels and transporters most often have unrelated structures, but there are rare examples of both existing within the same family of structurally similar proteins. CLC proteins, for example, include both chloride ion channels and transporters that pump chloride ions in one direction by harnessing the energy from hydrogen ions flowing in the other direction.

It remains unclear why some CLC proteins work as channels while others are transporters, especially since the two seem indistinguishable on the basis of the order of their amino acids – the building blocks of all proteins. The conservation of the amino acid sequences implies they are structurally very similar. How then can different members perform such energetically distinct processes?

Park and MacKinnon now show that the answer to this question serves as a reminder of how subtle nature can be. Indeed, while the structure of a human CLC channel (called CLC-1) is indeed similar to those of CLC transporters, one amino acid adopts a unique shape that explains why the protein cannot act as a transporter. This specific amino acid, a glutamate, is central to the exchange of chloride and hydrogen ions in CLC transporters. Park and MacKinnon show that its conformation in the CLC-1 channel stops this exchange, while leaving the pore open for the passive transport of chloride ions. Also, two other amino acids along the ion diffusion pathway in the CLC channel are smaller than their counterparts in CLC transporters, and so allow chloride ions to diffuse through more quickly. Lastly, Park and MacKinnon also note that channels do not require a wide pore: instead ions can still flow rapidly through a narrow pore if the chemical environment inside permits it.

CLC proteins perform a number of important roles in humans, and mutations in CLC-encoding genes underlie numerous heritable diseases. It remains too early to know how this mechanistic study may or may not impact treatments, yet the findings will likely interest scientists working on ion conduction mechanisms and the evolution of molecular function.

DOI: https://doi.org/10.7554/eLife.36629.002

---

these sites in various structures (*Dutzler et al., 2002*; *Dutzler et al., 2003*; *Feng et al., 2010*; *Jayaram et al., 2011*). In addition, the transporters all contain a glutamate residue positioned such that its side chain carboxylate group can bind either at $S_{ext}$ or $S_{cen}$ – in competition with a $Cl^-$ ion – or reside in the extracellular solution. Thus, CLC transporters are like $Cl^-$ channels with a weird feature – a glutamate side chain that clogs its own pore. This led to the idea that glutamate might not only be a competitor for the $Cl^-$ binding sites as the structures suggest, but it might also transfer a proton from inside to out (or the reverse) when it moves between its $S_{cen}$ position to its extracellular position (*Feng et al., 2010*; *Feng et al., 2012*). The transfer would naturally give rise to the 2:1 $Cl^-$:$H^+$ exchange stoichiometry characteristic of CLC transporters because 2 $Cl^-$ ions must be displaced when the glutamate gate moves between the extracellular solution and $S_{cen}$. This mechanism is consistent with the demonstrated conversion of a CLC transporter into a passive (but slow) $Cl^-$ channel upon mutation of the glutamate, as well as the demonstrated ability of small carboxylate-containing organic acids to compete with $Cl^-$ inside the pore (*Accardi et al., 2004*; *Accardi and Miller, 2004*; *Feng et al., 2012*). But there was one important caveat to make this transporter mechanism work: there must exist a relatively high kinetic barrier to $Cl^-$ flow near the intracellular side of the pore (*Feng et al., 2010*). This barrier would serve as the 'second gate' in the gating model conceptualization of transporters. So far, data for CLC transporters seem consistent with this mechanism: they have a channel-like pore, an external 'glutamate gate' that competes with $Cl^-$ binding and (presumably) transfers $H^+$ across the membrane, and structurally what appears to be a relatively high resistance (i.e., a large kinetic barrier) to $Cl^-$ flow near the intracellular aspect of the pore (i.e., the pore there is very narrow.)

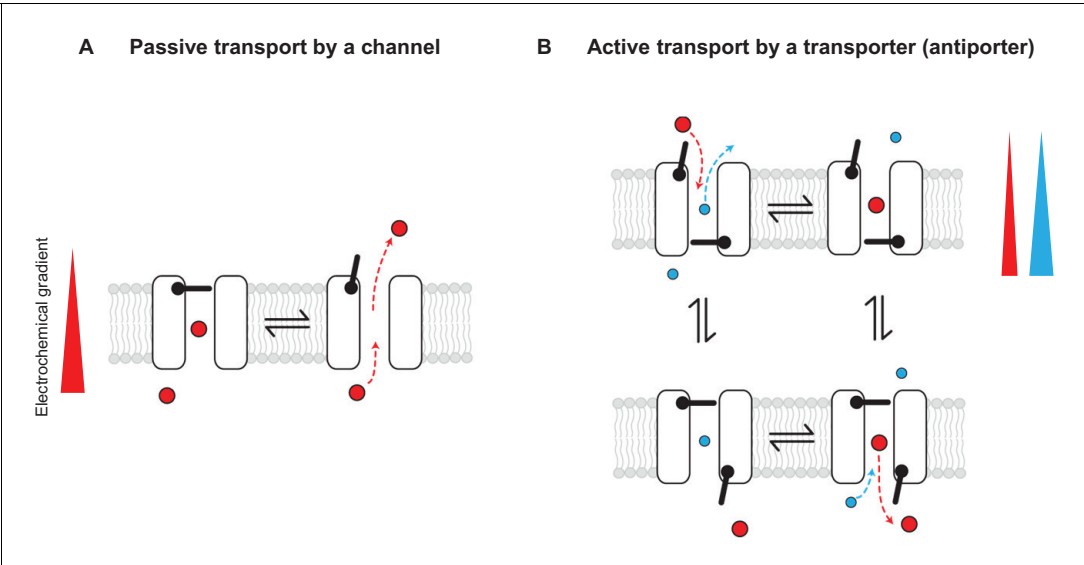

**Figure 1.** Passive and active transport explained by gating mechanisms. One-gate (A) and two-gate (B) models explaining passive transport by a channel and active transport by a transporter (shown is an antiporter). Direction of the solute electrochemical gradient is indicated by a wedge (the thicker end means more positive).

DOI: https://doi.org/10.7554/eLife.36629.003

Less is known about the chemistry and structure of CLC channels. Only one CLC channel structure has been determined, CLC-K from *Bos taurus* (referred to as bCLC-K or shortly CLC-K) (*Park et al., 2017*). This is a special case, a rare type of CLC channel that can be distinguished from CLC transporters based on its amino acid sequence because it does not have a 'glutamate gate'. That difference alone renders CLC-K inert to $H^+$ transfer. The structure of CLC-K also revealed a wider pore diameter on the intracellular side, consistent with a lowered kinetic barrier to $Cl^-$ flow. CLC-0/1/2 channels, by contrast, contain a 'glutamate gate' and are not distinguishable from CLC transporters by sequence. Thus, there must be an even more subtle distinction between these CLC channels and the transporters. Why does the glutamate gate in these channel CLCs not give rise to $H^+$ transfer coupled to $Cl^-$ transfer? Is a reduced kinetic barrier to $Cl^-$ flow near the intracellular side, suggested by the CLC-K structure, a common feature in CLC channels? To address these questions, we have determined the structure of CLC-1 from *Homo sapiens* (referred to as hCLC-1 or CLC-1).

We are also interested in the CLC-1 channel because it plays an important role in membrane repolarization of skeletal muscle cells following muscular contraction, and its mutation in humans causes hereditary muscle disorders known as *myotonia congenita* (*George et al., 1993*; *Koch et al., 1992*; *Lorenz et al., 1994*; *Steinmeyer et al., 1991*).

## Results

### Determination of a human CLC-1 channel structure by cryo-EM

We purified the CLC-1 protein in mild detergent from cultured human cells and examined them by cryo-EM single particle analysis (*Figure 2* and *Figure 2—figure supplements 1* and *2*). Despite its small molecular size (200 kDa), particles showed good contrast on micrographs under the optimized freezing and data acquisition conditions (*Figure 2A*). Two-dimensional (2D) class averages of selected particles displayed 2-fold rotational symmetry around an axis normal to the membrane (detergent micelle) (*Figure 2B*), as expected from the homodimeric architecture of CLC proteins (*Dutzler et al., 2002*; *Ludewig et al., 1996*; *Miller and White, 1984*). After removing artifacts and damaged particles by 2D classification, a density map was reconstructed at 3.9 Å resolution with C2 symmetry imposed (*Figure 2—figure supplement 1B*). This map showed a well-resolved transmembrane domain (TMD) with clearly visible α-helical features. By contrast, density for the carboxy-terminal cytosolic domain (CTD) was lower quality, suggesting conformational flexibility in this region.

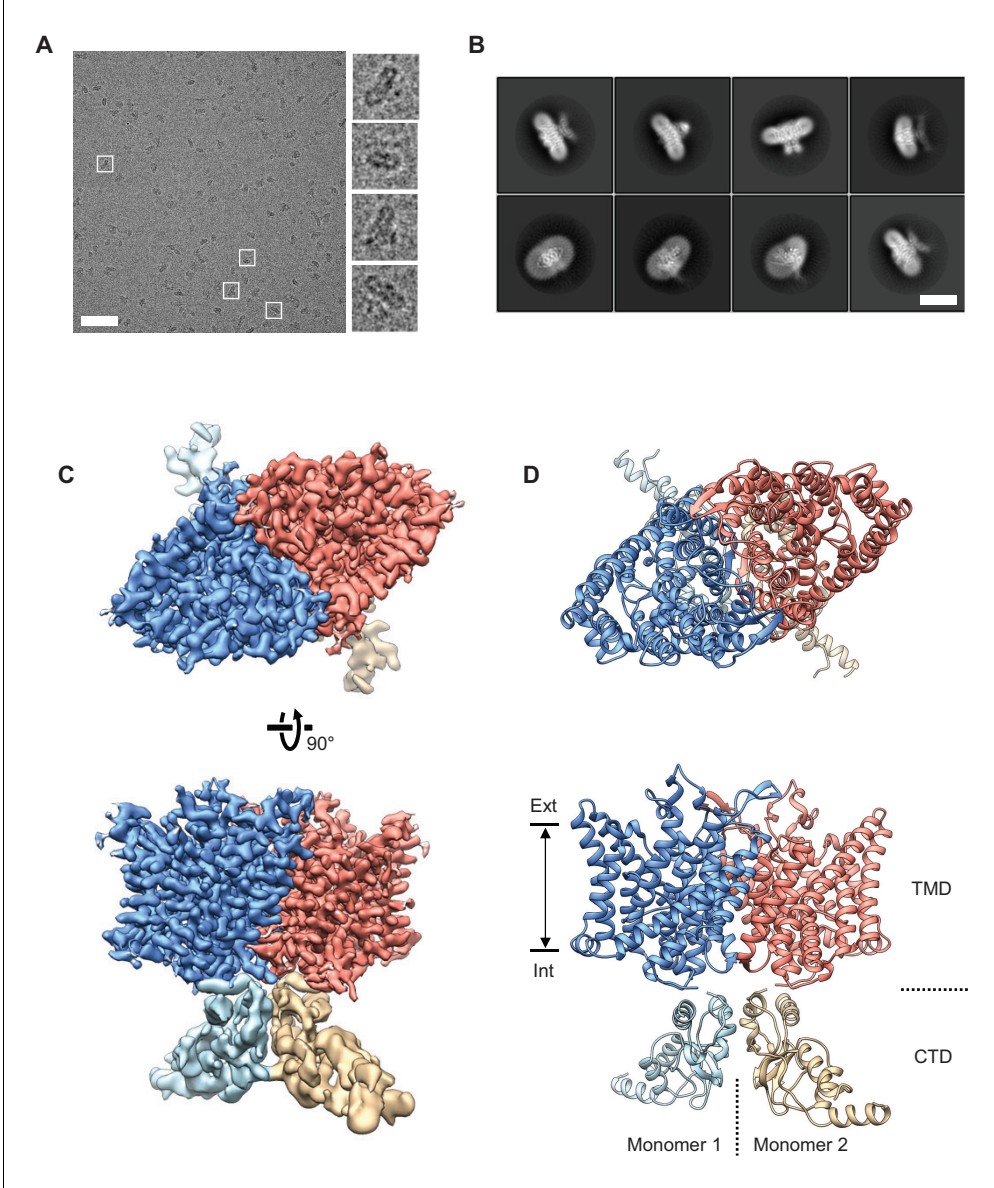

**Figure 2.** Cryo-EM structure of the human CLC-1 channel. (A) Representative micrograph of the purified CLC-1 channel (scale bar, 50 nm) on a cryo-EM grid. Representative particles (white squares) are magnified and shown in the right panels. (B) Images of selected 2D classes from reference-free 2D classification by RELION. Scale bar, 10 nm. (C and D) Cryo-EM density map (C) and atomic model (D) of the hCLC-1 channel. The transmembrane domain (TMD; blue and salmon) and the cytosolic domain (CTD; light blue and tan) were separately refined and combined for visualization. Ext, extracellular side. Int, intracellular side. The approximate lipid bilayer region is shown by arrows.

DOI: https://doi.org/10.7554/eLife.36629.004

The following figure supplements are available for figure 2:

**Figure supplement 1.** Sample preparation and cryo-EM image processing procedures.

DOI: https://doi.org/10.7554/eLife.36629.005

**Figure supplement 2.** Quality of the cryo-EM map and atomic model of CLC-1.

DOI: https://doi.org/10.7554/eLife.36629.006

To improve the map quality, we subjected particles to a round of 3D classification (*Figure 2—figure supplement 1B*). The results demonstrated that while the TMD is largely indistinguishable between classes, the CTDs deviate from each other by pivotal movements of varying degrees (*Figure 2—figure supplement 1C*). Based on this, we pooled ~170,000 particles from the two most populated and structurally similar classes, which correspond to 50% of particles. This particle set led to an improved density map at an overall resolution of 3.6 Å (data not shown). Using masking techniques to isolate individual regions, the resolution of the TMD was further improved to 3.4 Å (*Figure 2C*, *Figure 2—figure supplement 1B*, *Video 1*). The CTD remained poorly defined, likely due to continuous pivotal movements of its two wing-like structures (*Figure 2C*, *Figure 2—figure supplement 1C*, *Video 1*).

The good quality TMD density map enabled building a molecular model that included nearly all side chains (*Figure 2C*, *Figure 2—figure supplement 2*, and *Video 1*). The model was refined using Rosetta (*Wang et al., 2016*). The CTD map did not show side chain density but we could dock with confidence the crystal structure of the CLC-0 CTD (*Figure 2D* and *Figure 2—figure supplement 1D*) (*Meyer and Dutzler, 2006*). Both CLC-1 and CLC-0 channels contain a large loop extending from the CTD's cystathionin-β-synthase (CBS) domains, which was not visible in either the EM density map or the crystal structure. The function of the CTD is poorly understood; it may even be dispensable for ion transport given its high tolerance to mutation (*Estévez et al., 2004*) and absence in most bacterial CLC transporters.

## Bifurcated pore structure of CLC-1

The TMD of CLC-1 exhibits the canonical dimeric architecture of a CLC protein (*Figure 2C,D*). Each monomer is roughly a triangular prism shape and contains a complete ion transport pathway that appears structurally independent from that of the neighboring monomer. As in other CLC structures (*Dutzler et al., 2002*; *Dutzler et al., 2003*; *Feng et al., 2010*; *Park et al., 2017*), the $Cl^-$ transport pore in CLC-1 is most narrowly constricted halfway across the membrane, within the region referred to as the selectivity filter (*Figure 3A*). Overall, the pore lining is charged positive to attract $Cl^-$ (*Figure 3B*).

In contrast to other CLC proteins the potential route for ion diffusion in CLC-1 is bifurcated on the intracellular side of the selectivity filter– one following the canonical $Cl^-$ transport pathway found in all CLC proteins and the 'second' pore directed toward the protomer-protomer boundary on the cytosolic surface, which is distinctive in CLC-1 (*Figure 3A,B* and *Figure 3—figure supplement 1*). Both branches of the bifurcation are potentially hydrated because the radius is greater than that of water (1.4 Å) and the linings contain chemical groups with hydrogen bonding potential. A branch equivalent to CLC-1's secondary pore in the CLC-K channel is sealed off by F222 and V226 (corresponding to F288 and V292 of CLC-1) due to a different αH helix position (*Figure 3C*). In transporters, only a much narrower (~0.9–1.0 Å radius) pore could be detected, where stable dwelling of water molecules seems unlikely (*Figure 3—figure supplement 1*). In the *E. coli* transporter (EcCLC), the pore is further capped near the cytosolic surface by E203 (corresponding to V292 of CLC-1). We note that E203 of EcCLC and the equivalent Glu of mammalian CLC-4 and CLC-5 transporters have been implicated in shuttling $H^+$ between the intracellular solvent and the protein interior (*Lim and Miller, 2009*; *Lim et al., 2012*; *Zdebik et al., 2008*) by side-chain protonation and deprotonation, although this feature does not seem to be essential for $H^+$ transport in other cases, including the *C. merolae* transporter (CmCLC) (*Feng et al., 2010*; *Feng et al., 2012*; *Phillips et al., 2012*). It is possible that during $Cl^-/H^+$ exchange cycles, the αH helix of transporters transiently undergoes a conformational change such that a water-accessible pore is formed similarly to the CLC-1 case, which might facilitate $H^+$ transfer. Unlike transporter-type CLCs, the CLC-1 channel does not transport $H^+$

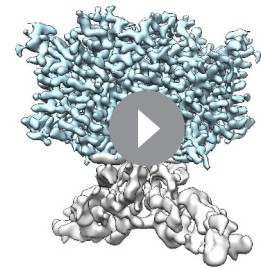

**Video 1.** Cryo-EM structure of the human CLC-1 channel. The cryo-EM map and atomic model of human CLC-1 are illustrated. Also see *Figure 2C and D*.

DOI: https://doi.org/10.7554/eLife.36629.007

in a manner tightly coupled to Cl$^-$ and thus it is unclear whether CLC-1's second intracellular pore is utilized for ion transport. Cl$^-$ ions may move through this pore in addition to the primary Cl$^-$ pathway.

## Chloride-selectivity filter and bound Cl$^-$ ions

The CLC-1 structure shows an anion selectivity filter largely similar to other CLC proteins but with some distinctive features (*Figure 4A* and *Video 2*). The filter is formed at the central constriction of the Cl$^-$ pathway by αN, αF, and αD helices, all of which point their N-terminal ends towards the center where Cl$^-$-binding sites are formed. This arrangement contributes to an electrostatically positive environment at the Cl$^-$-binding sites through α-helix end charges. Backbone nitrogen atoms from αN and αF segments are arranged to coordinate a partially dehydrated Cl$^-$ ion near the extracellular end of the constriction (external site or $S_{ext}$). In the CLC-1 density map we observe a clear density feature at $S_{ext}$, which likely corresponds to a bound Cl$^-$ ion (*Figure 4A* and *Video 2*). Typically, CLC proteins have two additional Cl$^-$-binding sites, namely, central ($S_{cen}$) and internal ($S_{int}$) sites (*Dutzler et al., 2003*). $S_{cen}$ has been observed to bind a Cl$^-$ ion through polar interactions with one or two backbone nitrogen atoms and the side chains of the conserved tyrosine (denoted $Tyr_C$; Y578 of CLC-1 or Y445 of EcCLC) and serine residues (denoted $Ser_C$; S189 of CLC-1 or S107 of EcCLC) (*Dutzler et al., 2002*; *Dutzler et al., 2003*). In the EcCLC transporter, $S_{cen}$ has been shown to bind Cl$^-$ relatively strongly (Kd ~1 mM) (*Lobet and Dutzler, 2006*; *Picollo et al., 2009*). $S_{int}$ is largely exposed to the intracellular solvent and binds Cl$^-$ with lower affinity ($K_d$ >20 mM) (*Lobet and Dutzler, 2006*; *Picollo et al., 2009*). In the CLC-1 map (determined in the presence of 116 mM Cl$^-$), $S_{int}$ shows a density peak whose intensity is comparable to that of the $S_{ext}$ density (*Figure 4A* and *Video 2*). By contrast, we do not observe density for an ion at $S_{cen}$ above the noise level, suggesting that $S_{cen}$ of CLC-1 may have a lower Cl$^-$ occupancy than $S_{ext}$ and $S_{int}$. This is somewhat surprising given the conservation of structural elements for $S_{cen}$, including $Tyr_C$ and $Ser_C$. Perhaps subtle structural differences account for the absence of an ion at this site compared to other CLC proteins. For example, we note that the position of $Tyr_C$ is shifted away from $S_{cen}$ by ~1.5 Å (see *Figure 5B*).

## New conformation of the gating glutamate

Like transporter-type CLC proteins and in contrast to CLC-K, the CLC-1 channel has a $Glu_{gate}$, but in CLC-1 it adopts a notably different conformation than previously observed in CLC transporters (*Figure 4* and *Figure 4—figure supplement 1*). Based on previous studies on transporters (*Dutzler et al., 2002*; *Dutzler et al., 2003*; *Feng et al., 2010*), $Glu_{gate}$, located in the immediate vicinity of $S_{ext}$ and $S_{cen}$, plays a key role in ion transport: when deprotonated its side-chain carboxylic moiety resides in either the $S_{ext}$ or $S_{cen}$ Cl$^-$ binding sites, preventing the binding of a Cl$^-$ ion therein. In the CLC-1 structure, the $Glu_{gate}$ side chain occupies neither $S_{ext}$ nor $S_{cen}$, but instead it is oriented in a different direction. The difference in $Glu_{gate}$'s conformation is mainly due to changes in its side-chain rotamer, whereas the polypeptide backbone arrangement in this region is similar among the structures (*Figure 4C*). The observed $Glu_{gate}$ conformation is also different than the outwardly-oriented (side chain projecting into the extracellular funnel) conformation that has been seen in the structure of an EcCLC Glu-to-Gln (E148Q) mutant (*Figure 4C*), which is hypothesized to mimic the protonated state of $Glu_{gate}$ (*Dutzler et al., 2003*).

It is unclear whether the $Glu_{gate}$ in the CLC-1 structure (determined at pH 7.4) is protonated. The pKa of the $Glu_{gate}$ side chain might be shifted towards a more neutral pH as it is neighbored by multiple hydrophobic amino acids (*Isom et al., 2010*). Yet, $Glu_{gate}$ at this position is more likely deprotonated because its side chain seems exposed to water molecules due to the presence of the second intracellular pore (*Figure 4B*). In CLC transporters, this conformation would be highly unfavorable because it would produce steric clashes with neighboring side chains (equivalent to V236, V265, and F279 of CLC-1; *Figure 4—figure supplement 1*), which are moved away in CLC-1 by a shift of the αG and αH helices. In other words, this conformation of $Glu_{gate}$ does not seem possible in CLC transporters studied so far.

The observed $Glu_{gate}$ conformation of CLC-1 was unexpected because it was never observed in other CLC protein structures, and yet it is consistent with an open CLC-1 channel, which is expected in the absence of an applied membrane potential. CLC-1 is a voltage-gated channel, which closes when the membrane potential is negative (i.e., at its 'resting' value) (*Fahlke et al., 1996*;

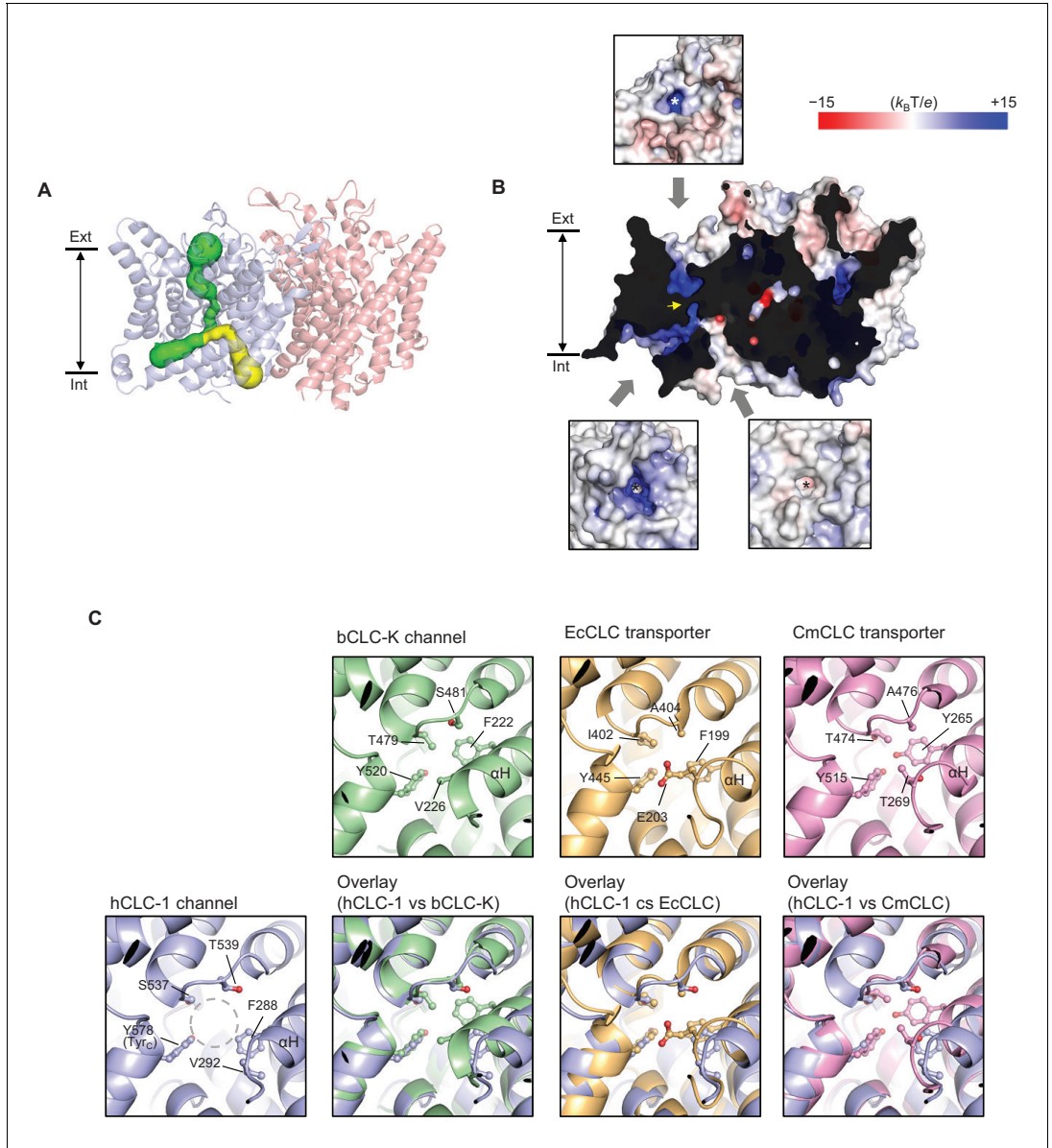

**Figure 3.** Bifurcated pore structure of the CLC-1 channel. (**A**) The canonical Cl⁻ transport pathway (green) and the second intracellular pore (yellow) are depicted in a side view of CLC-1. The pores of only one monomer are shown for simplicity. (**B**) Surface electrostatistics of CLC-1's pore lining. The protein surface was clipped to show optimally the pore lining of the CLC-1 monomer on the left. The yellow arrowhead indicates the position of the selectivity filter. The insets show views into the pore entrances, which are marked with asterisks. (**C**) A view into the second intracellular pore entrance from the cytosolic surface was compared to equivalent views with other CLC proteins. In the CLC-1 panel, the pore is indicated by a dashed gray circle. Amino acids lining the pore were indicated with their side chain atoms shown in ball-and-stick representation.

DOI: https://doi.org/10.7554/eLife.36629.008

The following figure supplement is available for figure 3:

**Figure supplement 1.** Comparison of the second intracellular pore among CLC proteins.

DOI: https://doi.org/10.7554/eLife.36629.009

*Pusch et al., 1995*). Perhaps in the presence of an applied negative membrane potential the $Glu_{gate}$ side chain moves into either the $S_{ext}$ or $S_{cen}$ position, as seen in CLC transporters, and prevents Cl⁻ conduction. This possibility would account for the observation that CLC-1 and related CLC-0 conduct Cl⁻ ions at all membrane voltages when the $Glu_{gate}$ residue is mutated to Gln (*Dutzler et al., 2003*; *Fahlke et al., 1997*).

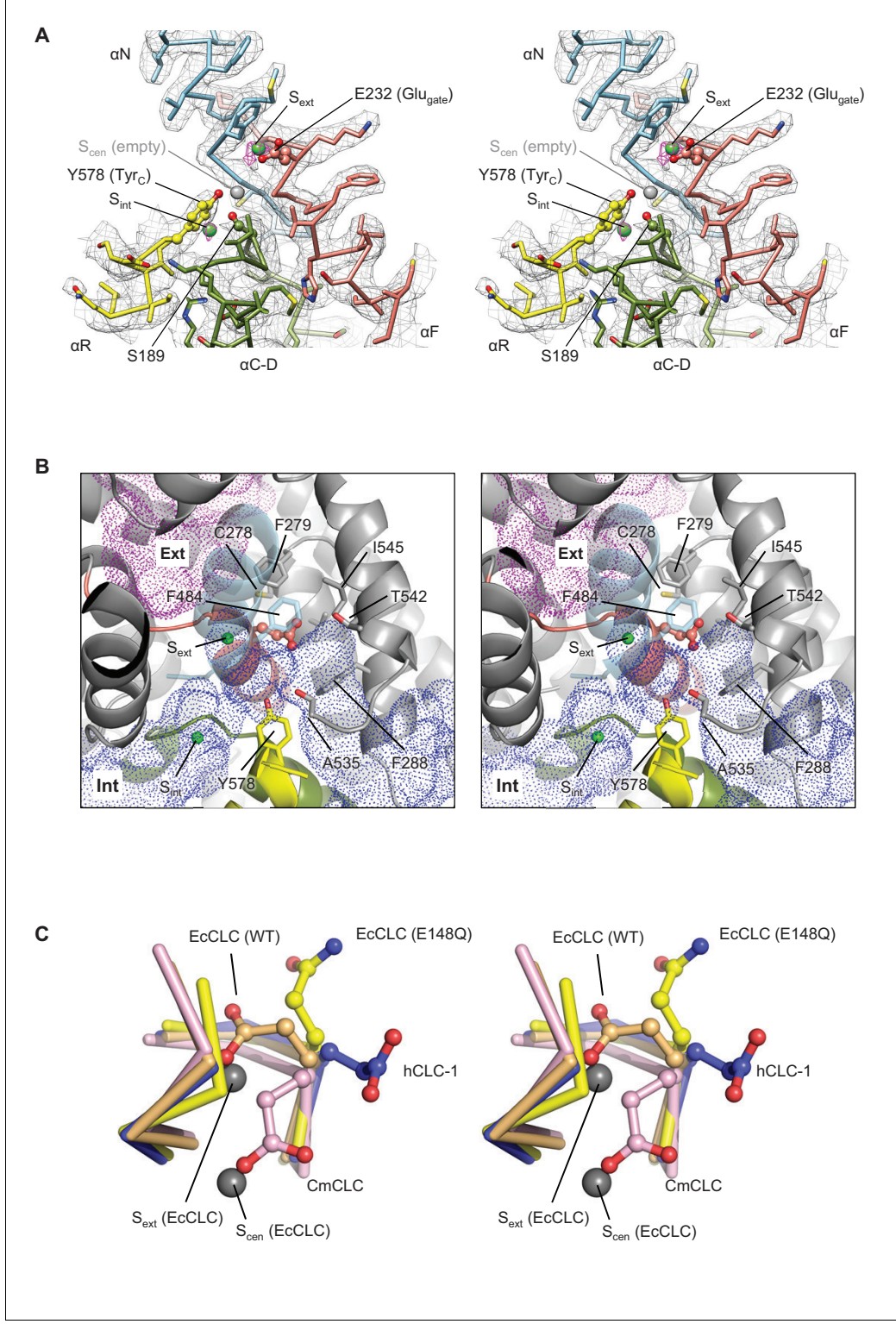

**Figure 4.** Glutamate gate (Glu_gate) and Cl⁻-binding sites of CLC-1. (**A**) View (stereo) into the selectivity filter of CLC-1. αN, αF, αR, and αC-D segments (Cα trace and side chains) are shown in cyan, salmon, yellow, and olive, respectively. The side chains of Glu_gate, Tyr_C, and Ser_C are represented with balls and sticks. Cl⁻-binding sites are indicated by green (S_ext and S_int) and gray (S_cen) spheres. The cryo-EM density map is shown in mesh (S_ext and S_int in magenta and the rest in gray). (**B**) Water-accessible regions in the filter region, probed by Hollow (*Ho and Gruswitz, 2008*), are shown with purple (extracellular vestibule) and blue (intracellular vestibule) dots. Glu_gate is

*Figure 4 continued on next page*

*Figure 4 continued*

represented in ball-and-stick. (**C**) Comparison of Glu$_{gate}$ positions between the CLC-1 channel and CLC transporters. The amino acid segments 146–149 and 355–358 forming the anion selectivity filter were aligned between structures. Cα-traces of the segments are shown with the Glu$_{gate}$ side chains in ball-and-stick representation. Blue, CLC-1. Light orange, WT EcCLC (PDB ID: 1OTS). Yellow, EcCLC E148Q mutant (PDB ID: 1OTU). Magenta, CmCLC (PDB ID: 3ORG). Gray spheres represent the positions of Cl$^-$ ions seen in EcCLC E148Q mutant (S$_{ext}$ and S$_{cen}$). Note that the Cl$^-$ ion at CLC-1's S$_{ext}$ (not shown) essentially coincides with S$_{ext}$ of EcCLC.
DOI: https://doi.org/10.7554/eLife.36629.010

The following figure supplement is available for figure 4:

**Figure supplement 1.** Comparison of the anion selectivity filter and Glu$_{gate}$ between CLC proteins.
DOI: https://doi.org/10.7554/eLife.36629.011

## 'Transporter-like' αC-D loop

The previous CLC-K channel structure has suggested that a wider pore diameter between S$_{cen}$ and S$_{int}$ is crucial for its channel function (**Park et al., 2017**). In CLC transporters, a kinetic barrier for Cl$^-$ passage (i.e., a narrowing of the pore) exists on the intracellular side of the vestibule to preclude slippage of Cl$^-$ ions during the Cl$^-$/H$^+$ exchange cycle (**Feng et al., 2010**). This barrier is due to a narrow pore width between S$_{cen}$ and S$_{int}$, which is created in part by Ser$_C$ of the αC-D loop interposed between the two Cl$^-$ binding sites. In the CLC-K structure, the αC-D loop has a distinctly different conformation, where Ser$_C$ is flipped down and thus no longer interposed between the two Cl$^-$ binding sites. Consequently, the pore diameter is wider such that Cl$^-$ ions will more readily permeate. Given that CLC-1 is also a channel, we wondered whether the αC-D loop in CLC-1 would adopt a similar 'flipped-down' conformation.

While a different conformation of the αC-D loop is a key feature distinguishing CLC-K from transporters, a structural comparison shows that this is not the case for the CLC-1 channel (**Figure 5**). In contrast to CLC-K, the αC-D loop in CLC-1 adopts the loop conformation seen in CLC transporters, especially CmCLC (**Feng et al., 2010**). Consequently, the Ser$_C$ side chain is positioned between S$_{int}$ and S$_{cen}$ (**Figure 5B**). Therefore, in the case of CLC-1 the αC-D loop itself does not provide an explanation for why CLC-1 functions as a Cl$^-$ channel (see below). This also suggests that the 'flipped-down' conformation of Ser$_C$ may be unique to the CLC-K channel.

## Comparison of Cl$^-$ pore structures of CLC proteins

To understand why CLC-1 functions as a channel we compared its Cl$^-$ pore structure to that of other CLC proteins. In both CLC-1 and CLC-K channels, a continuous Cl$^-$ pathway was evident in between the extracellular and intracellular funnels, through the selectivity filter (**Figure 6A,B**). In the EcCLC and CmCLC transporters, a continuous pore could be detected only when the Glu$_{gate}$ side-chain atoms (from Cβ) were excluded from the pore radius calculation as Glu$_{gate}$ sits at S$_{ext}$ or S$_{cen}$ (**Figure 6C,D**). These results would therefore reflect the pore structure when the transporter's Glu$_{gate}$ transiently moves away from the Cl$^-$ pathway upon protonation (hypothetically, akin to the crystal structure of the EcCLC E148Q mutant). However, we note that calculated pore radii around S$_{ext}$ may be somewhat overestimated due to the actual presence of the Glu$_{gate}$ side-chain atoms.

The CLC-1 channel has the narrowest (1.0 Å in radius) constriction above S$_{ext}$ toward the extracellular side due to the placement of the M485 side chain near the external end of the Cl$^-$ pathway (**Figure 6A**). While the radius is

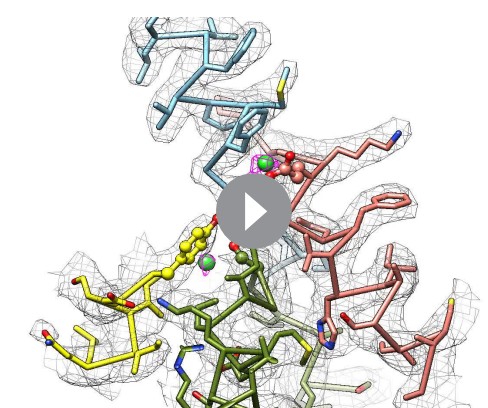

**Video 2.** Glutamate gate (Glu$_{gate}$) and Cl$^-$-binding sites of CLC-1. The selectivity filter region of human CLC-1 is shown. The same color scheme and representation are used in **Figure 4A**.
DOI: https://doi.org/10.7554/eLife.36629.013

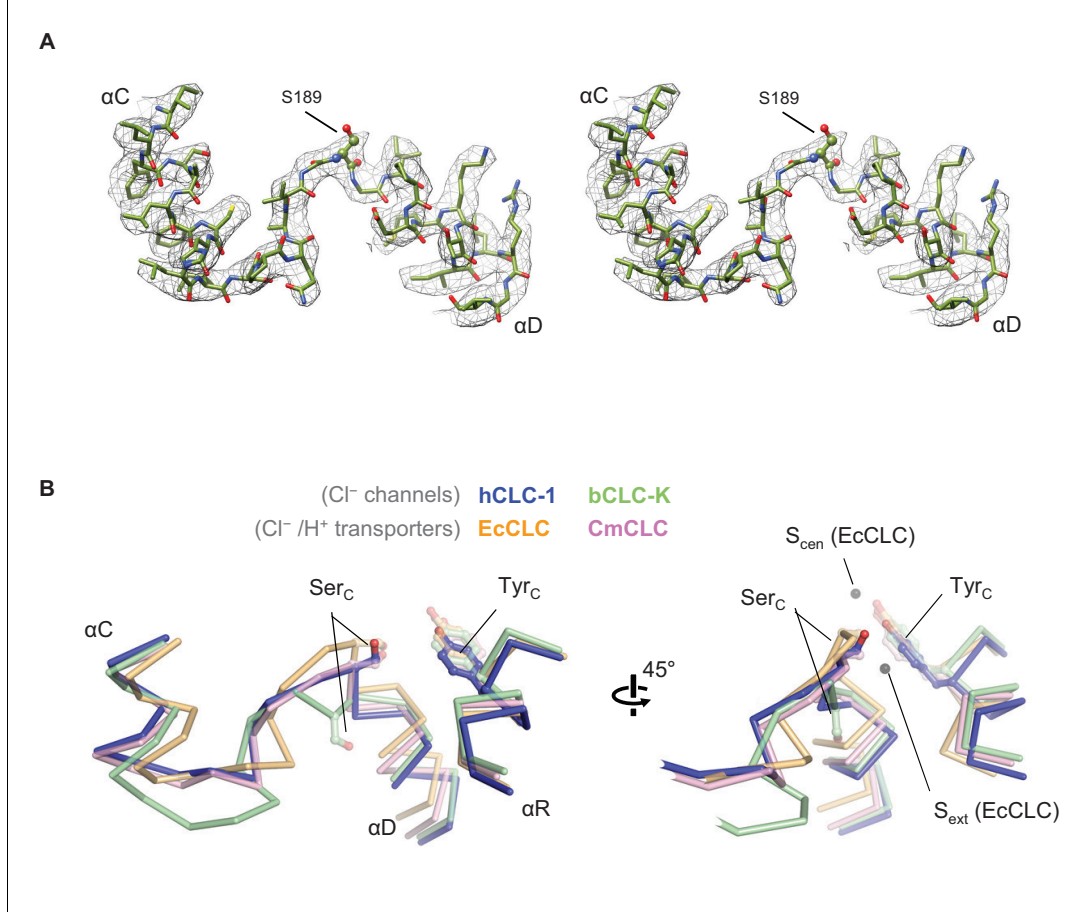

**Figure 5.** Structure of the αC-D loop and Ser$_C$. (**A**) Atomic model and cryo-EM density of the CLC-1 αC-D. (**B**) Comparison of the αC-D and αR segments (shown in Cα-only trace) among CLC channels and transporters. The side chains of Ser$_C$ and Tyr$_C$ are shown in sticks.
DOI: https://doi.org/10.7554/eLife.36629.012

significantly smaller than the Cl$^-$ radius (~1.7 Å), the flexibility of the M485 side chain must allow Cl$^-$ ions to pass through this region. Given the narrowness of this constriction, it is likely that M485 affects the Cl$^-$ throughput of the channel. In fact, its mutation to less flexible valine (M485V) causes recessive myotonia congenita and has been shown to reduce the single channel conductance of CLC-1 to about 20% of the wild type channel conductance (*Wollnik et al., 1997*).

From S$_{ext}$ to the intracellular opening the CLC-1 channel structure shows a relatively wide pore opening despite its 'transporter-like' αC-D loop. This suggests the absence of a large kinetic barrier in CLC-1, but for reasons other than the αC-D loop conformation. Compared to the CLC-K channel, CLC-1 has a slightly narrower (1.5 Å vs 1.7 Å in radius) opening between S$_{cen}$ and S$_{int}$ because of Ser$_C$. This might create a kinetic barrier to some degree, but the pore is still significantly wider and more hydrophilic than the equivalent region in the EcCLC transporter (*Figure 6C*). The difference originates mainly from two amino acids (T475 and G483) lining the constriction. In EcCLC, the equivalent positions are F348 and I356, which project their bulky, hydrophobic side chains towards the Cl$^-$ pathway between S$_{cen}$ and S$_{int}$. Together with proximal placement of Ser$_C$ and Tyr$_C$, this narrows the opening (1.0 Å in radius) in EcCLC. In the CmCLC transporter, the constriction at the kinetic barrier region is wider (1.6 Å in radius) than EcCLC because of smaller side chains at the equivalent positions (I421 and V429; *Figure 6D*) and a slight downward shift (1.5 Å) of Ser$_C$ with respect to the positions in EcCLC (*Figure 5B*). Yet, hydrophobicity provided by the I421 and V429 side chains might result in a significantly higher kinetic barrier than in CLC-1.

It is noteworthy that the CLC-1 channel shows a 1.5 Å outward shift of the Tyr$_C$ side chain with respect to the position that is almost invariant in the other CLC structures (*Figure 5B*). In CLC-1, this

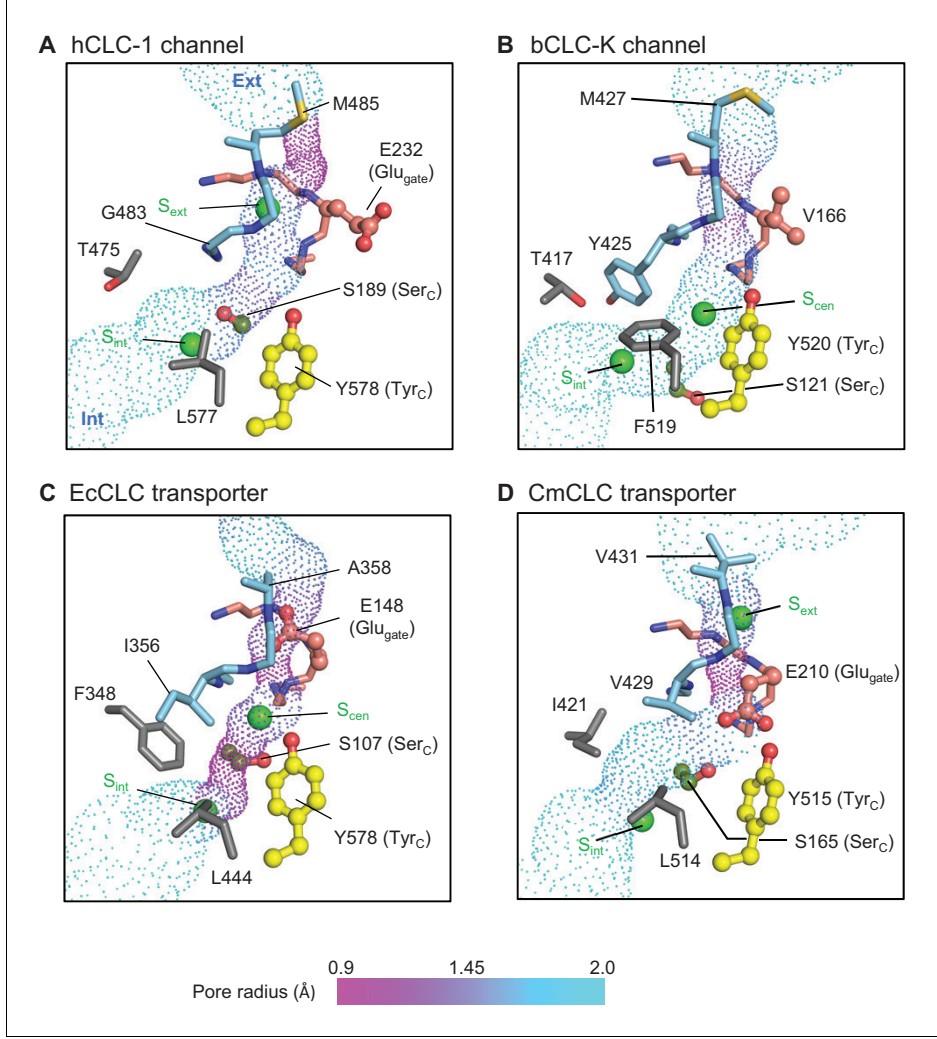

**Figure 6.** Profile of the Cl⁻ transport pore and the kinetic barrier between $S_{cen}$ and $S_{int}$. Pore structures along the Cl⁻ pathway are shown in dot representation together with amino acids around it. $Glu_{gate}$, $Tyr_C$, and $Ser_C$ side chains are shown in ball-and-stick representation. The color scheme is the same as in *Figure 4*. Pore-lining amino acids that are distinctive between CLC channels and transporters are shown in gray.
DOI: https://doi.org/10.7554/eLife.36629.014

The following figure supplement is available for figure 6:

**Figure supplement 1.** Sequence comparison of selected amino acids forming the ion selectivity filter and the pore lining.
DOI: https://doi.org/10.7554/eLife.36629.015

shift contributes to pore widening in the cytosolic vestibule. At present it is unclear if this shift of $Tyr_C$ is static or part of dynamic movements in CLC-1 and if it is unique in CLC-1 or a similar movement exists in other CLC proteins. Previous biophysical studies have proposed a movement of $Tyr_C$ to explain alternating gate opening of the EcCLC Cl⁻/H⁺ transporter (*Basilio et al., 2014*; *Jayaram et al., 2008*; *Khantwal et al., 2016*). On the other hand, EcCLC crystal structures obtained with a number of different variants and crystallization conditions have not yet revealed any movement of $Tyr_C$.

## Distinctive amino acid pattern between CLC channels and transporters around the kinetic barrier region

Because the CLC-1 structure suggests that T475 and G483 (equivalent to F348 and I356 in EcCLC, respectively) likely contribute to lowering of the kinetic barrier, we compared amino acids lining this

region among both CLC channels and transporters (*Figure 6—figure supplement 1*). Indeed, these two positions showed a distinctive differential pattern when comparing CLC channels and transporters, whereas other positions (i.e., H369, C481, L577, and I581 in CLC-1) did not. Generally, these two positions are filled with large, hydrophobic amino acids in transporters but are replaced by a small, polar amino acid in CLC channels. One notable outlier is position 417 of the CLC-K channels (Y425). However, the CLC-K channel structure shows that its phenyl side chain is skewed off the $Cl^-$ pathway, and thus does not seem to create a kinetic barrier in CLC-K (*Figure 6B*). In fact, it forms the $S_{cen}$ $Cl^-$ binding site together with $Tyr_C$ and F519 through anion-quadrupole interactions (*Park et al., 2017*) (*Figure 6B*). In summary, the observed amino acid pattern and structural information suggest that a lowered barrier in the $S_{cen}$–$S_{int}$ region of the pore is a common feature of CLC channels, but CLC-1 and CLC-K channels achieve this somewhat differently. In the CLC-1 channel, small side chains in pore-lining residues lower the kinetic barrier, whereas in CLC-K mainly the reorientation of $Ser_C$ lowers it. The extent of the kinetic barrier should also be affected by the hydrophobic and electrostatic nature of the lining residues, not only the physical dimensions of the pore.

## Working model and experimental validation

Combining the new structural information and previous data, we propose a working model that channel behavior in CLC proteins arises out of the following physical conditions (*Figure 7*): (1) Glu$_{gate}$ is either absent (i.e., in CLC-K) or allowed to reside in an 'open' configuration (i.e., CLC-1) for a sufficiently extended period of time (rather than occupying $S_{ext}$ or $S_{cen}$); (2) a lowered kinetic barrier between $S_{cen}$ and $S_{int}$; (3) reduced $Cl^-$-binding affinity at $S_{cen}$ (or $S_{ext}$, as suggested by apparent low occupancy in the CLC-K structure). A reduced kinetic barrier would be an important feature to achieve fast $Cl^-$ throughput. On the other hand, a sufficient kinetic barrier would be crucial in transporters to preclude undesired slippage of $Cl^-$ ions through the transiently open pore (*Feng et al., 2010*). In addition, reduction of $Cl^-$-binding affinity at $S_{cen}$ and/or $S_{ext}$, which is energetically related to the kinetic barrier, might also contribute to high $Cl^-$ throughput in channels. For example, relatively deep energy wells at $S_{cen}$ and $S_{ext}$, as implied by the high occupancy of sites in the EcCLC transporter, would create a larger energy difference between the binding sites and the 'transition states', which effectively raises the energy barrier. In CLC-1 the relatively low binding site occupancy implies not very deep energy wells and thus a smaller energy difference between the binding sites and 'transition states'.

We carried out biophysical experiments to test some of these ideas using the EcCLC transporter (*Figure 8A,D*). EcCLC mutants were produced, purified and reconstituted into lipid vesicles for assessment of $Cl^-$ and $H^+$ transport activity (*Figure 8* and *Figure 8—figure supplement 1*) (*Feng et al., 2012*; *Jayaram et al., 2008*; *Walden et al., 2007*). The ideas outlined above predict that if the kinetic barrier in EcCLC is lowered it should behave more like a CLC channel (i.e., rapid $Cl^-$ permeation with decreased $H^+$ transport activity). $Cl^-$ permeation is expected to be further increased if the Glu$_{gate}$ is rendered persistently opened. As reported previously (*Jayaram et al., 2008*), opening of Glu$_{gate}$ alone by the Glu-to-Ala mutation (E148A) abolishes the $H^+$ transport activity, but it also reduced $Cl^-$ throughput by a factor of approximately 0.25. We reason that this is likely because the mutant still retains the kinetic barrier deterring $Cl^-$ ions from moving between $S_{cen}$ and $S_{int}$. Thus, while removal of the Glu$_{gate}$ is sufficient to convert the transporter into a $Cl^-$ channel, a reduced kinetic barrier would be key to an increased $Cl^-$ throughput, an important feature of the native CLC channels.

Previous studies have shown that when the E148A mutation is combined with a $Tyr_C$ mutation (e. g., Y445S), the $Cl^-$ transport rate dramatically increases (*Jayaram et al., 2008*), demonstrating that efficient $Cl^-$ channel activity can be produced from EcCLC by altering its gates. However, we note that Y445S is rather unphysiological as $Tyr_C$ is invariant among all CLC channels and transporters. Therefore, here we examined the effects of lowering the kinetic barrier in wild type EcCLC by mutating $Ser_C$ or neighboring pore-lining amino acids, guided by the CLC-1 and CLC-K structures (*Figure 8B,E* and *Figure 8—figure supplement 1*). Trimming the side chain of $Ser_C$ (S107G), with the intention of mimicking the flipped $Ser_C$ in the CLC-K channel, increased the $Cl^-$ transport rate by a factor of 2, as previously reported (*Jayaram et al., 2008*). At the same time, this mutation lowered $H^+$ coupling 3-fold, as one would expect due to the slippage of uncoupled $Cl^-$ ions. Next, since CLC-K has a polar amino acid (Thr) at one of its pore-lining residues (F348 of EcCLC), we further introduced a similar (F348A) mutation. This increased the $Cl^-$ throughput and almost abolished

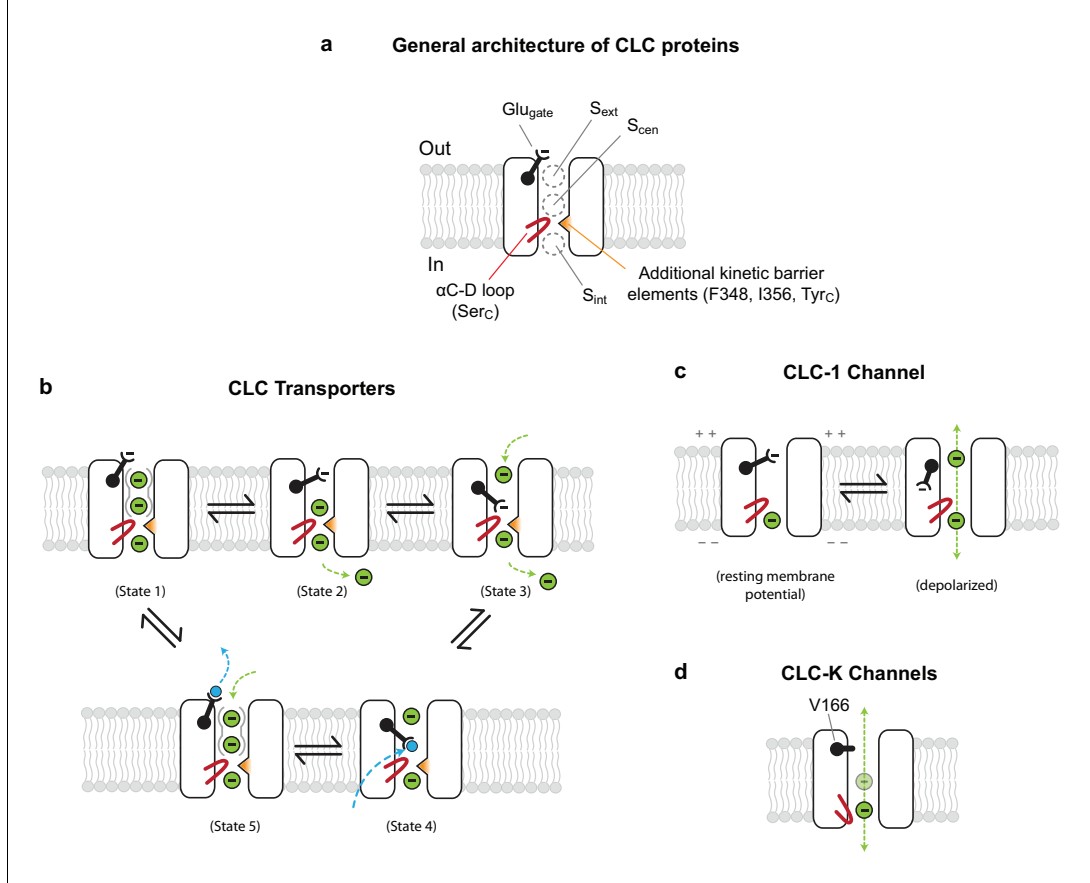

**Figure 7.** Models for ion transport mechanisms of CLC transporters and channels. (**A**) General architecture of CLC proteins. F348 and I356 are labelled according to *E. coli* transporter (EcCLC) numbering. (**B**) Model for 2:1 $Cl^-/H^+$ exchange by CLC transporters. The negatively-charged carboxylic group of the $Glu_{gate}$ side chain can occupy $S_{ext}$ (state 2) or $S_{cen}$ (state 3) by a swinging motion, competing with a $Cl^-$ ion for binding therein. When protonated at $S_{cen}$ by a proton transferred from the cytosol (state 4), the $Glu_{gate}$ side chain flips out to the extracellular side (state5). The kinetic barrier between $S_{cen}$ and $S_{int}$ would prevent leakage of $Cl^-$ ions through the open pore during this transient step. Also, as seen previously with EcCLC (***Picollo et al., 2009***), synergistic binding of two $Cl^-$ ions at $S_{ext}$ and $S_{cen}$ (depicted by solid gray curves around $S_{ext}$ and $S_{cen}$) might further deter slippage of $Cl^-$ ions. Deprotonation resets the cycle (state1). The cycle is reversible, and for simplicity the intermediate steps were omitted. $Cl^-$ ions and $H^+$ are depicted as green and blue spheres, respectively. (**C**) Model for the CLC-1 channel. The cryo-EM structure of CLC-1 presented in this study represents the depolarized state. Although the conformation of the $\alpha$C-D loop remains similar to that of transporters, CLC-1's kinetic barrier is lower than transporters due to the lack of additional kinetic barrier elements. In addition, weak $Cl^-$-binding affinity at $S_{cen}$ might facilitate rapid permeation of $Cl^-$ ions along the pore. When the membrane potential is negative (resting), the $Glu_{gate}$ side chain may occupy $S_{ext}$ or $S_{cen}$ as in transporters, blocking the pore. (**D**) Model for CLC-K channels. The outer gate is removed by a natural mutation of $Glu_{gate}$ to valine (V166). The kinetic barrier is largely reduced due to a flip-down of $Ser_C$, as well as lack of other kinetic barrier elements. The cryo-EM structure suggested that $S_{ext}$ and $S_{cen}$ have weaker $Cl^-$-binding affinity than transporters (empty and with a semi-transparent $Cl^-$ sphere).

DOI: https://doi.org/10.7554/eLife.36629.016

coupled $H^+$ transport. Finally, by adding a $Glu_{gate}$ mutation (E148A) to this double mutant the $Cl^-$ throughput was further increased. Compared to the E148A single mutant, the triple mutant (S107G/F348A/E148A) has a $Cl^-$ transport rate increased about 25-fold (***Figure 8—figure supplement 1***).

Similar results were obtained when mutations mimicking the CLC-1 channel were introduced to EcCLC (***Figure 8C,F*** and ***Figure 8—figure supplement 1***). While single mutations at the pore-lining amino acids (F348T or I356G) did not increase the $Cl^-$ transport rate, the double mutation (F348T/I356G) moderately increased the $Cl^-$ throughput (1.5-fold with respect to the wildtype). We note that this mutant displayed no measurable $H^+$ transport activity. When the double mutant was combined with the $Glu_{gate}$ mutation (E148A), which was used as a surrogate of the $Glu_{gate}$ conformation observed in the CLC-1 structure, the $Cl^-$ throughput dramatically increased (22-fold with respect to the single E148A mutant; ***Figure 8—figure supplement 1***). Single mutations (F348T or I356G) in the

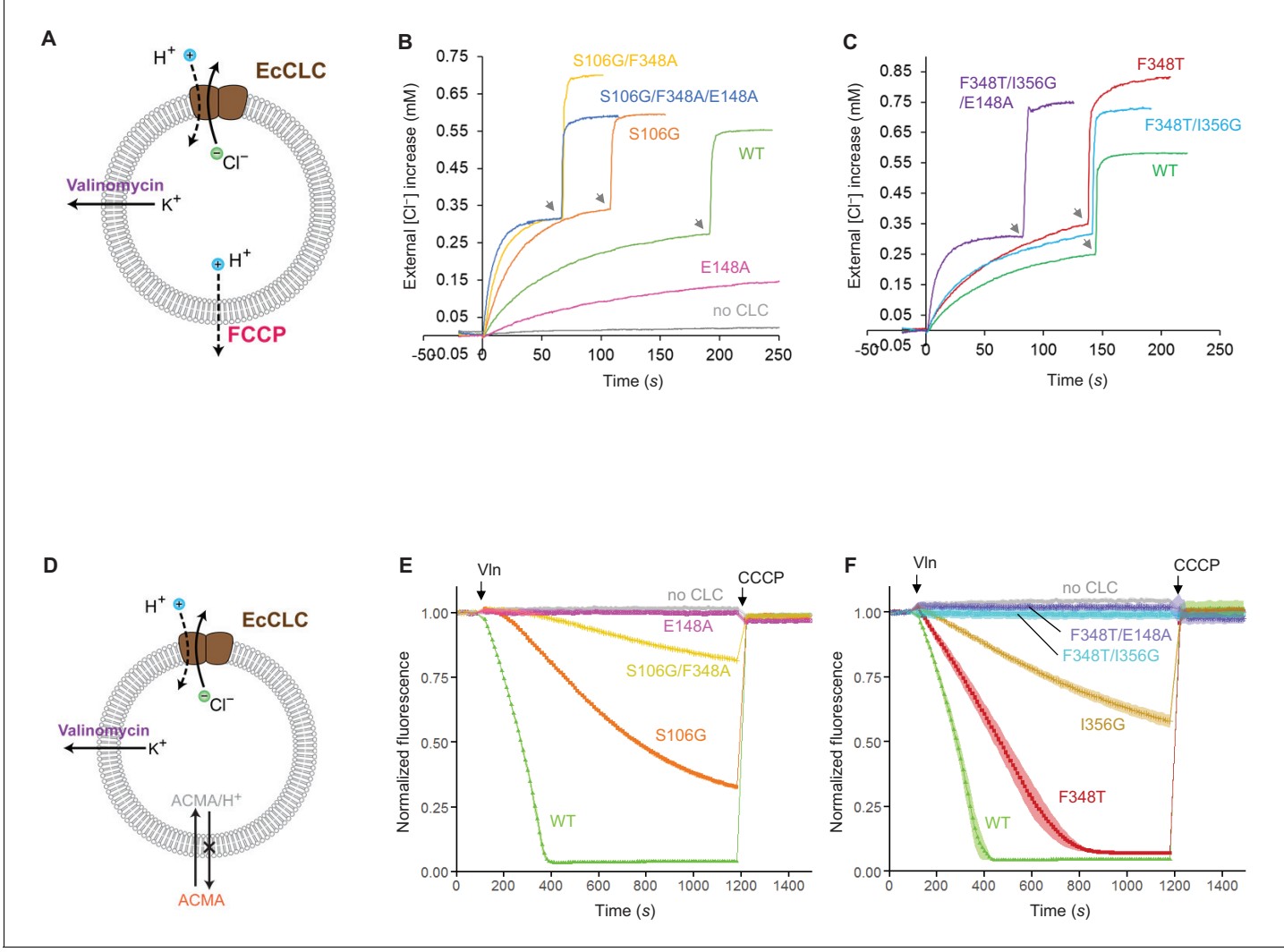

**Figure 8.** Effects of Glu$_{gate}$ and kinetic barrier mutations on Cl$^-$ and H$^+$ transport activities of EcCLC. (A) Schematics of the Cl$^-$ dump assay to measure the Cl$^-$ transport rate of EcCLC. Purified EcCLC protein is reconstituted into proteoliposomes containing 300 mM KCl inside. Buffer outside the liposomes was reduced to 150 mM K$_2$SO$_4$, lowering the Cl$^-$ concentration outside to ~1 mM. Transmembrane ion flux was initiated by addition of the K$^+$-ionophore valinomycin (Vln) and the protonophore carbonyl cyanide-4-(trifluoromethoxy)phenylhydrazone (FCCP). Increase of the Cl$^-$ concentration outside the liposomes was monitored using a Cl$^-$-selective electrode. (B) and (C) Examples of raw traces of Cl$^-$ dump assays. Vln/FCCP was added at t = 0. the gray arrowheads indicate addition of the β-octyl glucoside detergent to the reaction to release all Cl$^-$ from liposomes. (D) Schematics of the fluorescence-based H$^+$ influx assay to measure the H$^+$ transport activity of EcCLC. EcCLC proteoliposomes containing 450 mM KCl inside were diluted to buffer containing 450 mM potassium gluconate, lowering Cl$^-$ concentration outside to ~30 mM. The flux was initiated by addition of valinomycin at t = 100 s. As H$^+$ are transported into the vesicles by EcCLC, intravesicular pH drops, which can be monitored by the quenching of 9-amino-6-chloro-2-methoxyacridine (ACMA) fluorescence. At the end of experiments (t = 1200 s), the protonophore carbonyl cyanide m-chlorophenylhydrazone (CCCP) was added to release all H$^+$ from the vesicles. (E) and (F) Examples of normalized fluorescence traces of ACMA-based H$^+$ influx assay. Shown are means (line and symbols) and s.e.m. (band) of 4 experiments.

DOI: https://doi.org/10.7554/eLife.36629.017

The following figure supplement is available for figure 8:

**Figure supplement 1.** Summary of relative Cl$^-$ and H$^+$ transport activities of EcCLC mutants.

DOI: https://doi.org/10.7554/eLife.36629.018

E148A background showed intermediate increases in Cl$^-$ throughput, suggesting that the effects of these mutations are somewhat additive.

# Discussion

The human CLC-1 channel exhibits interesting structural differences in the $Cl^-$ transport pathway and the gates, which can explain why this protein functions as a $Cl^-$ channel instead of a $Cl^-/H^+$ antiporter. The outer gate of the channel remains open because the carboxylic side-chain $Glu_{gate}$ is located off to the side, away from the $Cl^-$ transport pathway (*Figure 4*). The inner kinetic barrier seems to be substantially lowered compared to transporters owing to a wider pore diameter near the cytosolic side (*Figure 6*). The pore widening is subtle, but distinctive enough to reveal a pattern separating channels and transporters at the protein sequence level (independent of the presence or absence of a $Glu_{gate}$) (*Figure 6—figure supplement 1*).

The position of the $Glu_{gate}$ residue in CLC-1 is unique among CLC structures so far observed. The new $Glu_{gate}$ position, where its carboxylic side chain is directed off to the side of the $Cl^-$ pathway, is enabled by a pocket that is large and hydrophilic (owing to its bifurcated pore structure) enough to accommodate $Glu_{gate}$'s side chain. This pocket may also exist in other $Glu_{gate}$-containing CLC channels (i.e., CLC-0 and CLC-2) but does not seem to exist in transporters because of a different arrangement of neighboring amino acids. It seems likely that this $Glu_{gate}$ position is key to understanding why CLC-1 exhibits a stable open (i.e., conducting) state. On the basis of mutagenesis studies (*Dutzler et al., 2003*; *Fahlke et al., 1997*), the $Glu_{gate}$ in CLC-0 and CLC-1 has been identified as a 'voltage sensor' because its removal abolishes voltage-dependent gating. From this observation, we would suggest that the position of $Glu_{gate}$ (i.e., whether it resides off to the side, not occluding the pore, or within the pore) depends on the transmembrane voltage and generally dictates gating each CLC-1 monomer's pore (also referred to as a 'protopore').

An unresolved issue raised by the new $Glu_{gate}$ side chain conformation is this: if this conformation corresponds to the conducting state, how is it favored by low pH outside (*Rychkov et al., 1996*)? One possibility is the $Glu_{gate}$ might be protonated in this conformation. Alternatively, low pH might stabilize a conformation of $Glu_{gate}$ outside the pore, as in the EcCLC E148Q mutant. This conformation would also remove $Glu_{gate}$ from the pore and permit conduction. Finally, the pH effect might be produced allosterically by protonation of an unidentified amino acid on the extracellular side. For example, both CLC-2 and CLC-K channels are inhibited by external pH <6.5, but it has been shown that a His residue (H532 of CLC-2 and H497 of CLC-K), which is located ~20 Å away from the pore, is responsible for this effect (*Gradogna et al., 2010*; *Niemeyer et al., 2009*). This issue remains unresolved for now.

Functional experiments using EcCLC provide support for our model that a low kinetic barrier in the cytosolic vestibule is necessary for high $Cl^-$ transport rates, which are general characteristics of native CLC channels (*Figure 8* and also see (*Jayaram et al., 2008*)). The results indicate that a small increase in the pore diameter and a decrease in hydrophobicity of the pore lining can substantially lower the kinetic barrier. The structures, however, suggest that the extent might be somewhat less in the CLC-1 channel than in the CLC-K channel because of CLC-1's $Ser_C$ 'transporter-like' conformation. This is in fact consistent with the observation that CLC-1 has vestigial $H^+$ transport activity (*Picollo and Pusch, 2005*) and a relatively slow $Cl^-$ throughput compared to that of CLC-K channels (1.2–1.8 pS versus 20–30 pS of CLC-K) (*L'Hoste et al., 2013*; *Saviane et al., 1999*; *Scholl et al., 2006*; *Weinreich and Jentsch, 2001*). What then causes the $Ser_C$ to adopt its flipped-down conformation in the CLC-K channel? In CLC-K, position 425 contains a bulky amino acid (Y425), in contrast to other CLC proteins. In the canonical conformation $Ser_C$ would sterically clash with Y425 (e.g., the center-to-center distance between the $Ser_C$-Oγ and Y425-Cε atoms would become 2.3 Å). We speculate that this steric incompatibility imposed by the unique Y425 might lead to the flipped-down conformation of $Ser_C$ in the CLC-K channel.

The observed low $Cl^-$ occupancy at $S_{cen}$ in the CLC-1 structure has a striking resemblance to previous crystallographic observations on EcCLC, wherein $S_{cen}$ remained unoccupied when experiments were performed with $Tyr_C$ mutants lacking $H^+$ transport activity or with pseudohalides, which permeate without coupled $H^+$ transport (*Accardi et al., 2006*; *Nguitragool and Miller, 2006*). It has been shown that in EcCLC, low $Cl^-$ occupancy correlates with low anion binding affinity (*Picollo et al., 2009*). This comparison suggests a reduced $Cl^-$ binding affinity at $S_{cen}$ in CLC-1, although further biophysical measurements will be necessary to confirm this. We speculate that this feature contributes to reduced $H^+$ transport and increased $Cl^-$ conduction. Possible causes underlying the altered $Cl^-$ affinity include the shifted position of $Tyr_C$ and subtle changes in positions and orientations of

neighboring backbone nitrogen atoms coordinating the Cl⁻ ion. For example, we note that CLC-0/1/2 channels have smaller, more flexible residues (Gly or Ala) at the G483 position, in contrast to Leu, Ile, or Val in CLC transporters.

CLC-1 is now the second structure of a channel-forming CLC, the first being CLC-K (*Park et al., 2017*). One of the major features giving rise to channel behavior is a more conductive pore. The structural differences giving rise to the higher Cl⁻ conductivity are fairly subtle: the pore is slightly wider and the chemical properties a little different, accounting for what we propose to be a reduced kinetic barrier. We think there is a very important lesson here. Throughput rates in the range of $10^6$ ions per second do not require a wide pore. We conclude that even if the pore in places is on average narrower than the ion, as long as the lining atoms are favorable to a conducting ion with respect to their electrostatic and chemical properties, and as long as they are sufficiently dynamic (*i.e.* they can move out of the way), then the ion can diffuse through. We offer as an example of this idea, the selectivity filter of $K^+$ channels (*Zhou et al., 2001*). The atomic structures show us that in fact the pore's radius between the $K^+$ binding sites is smaller than the radius of a $K^+$ ion. And yet some $K^+$ channels approach throughput rates of $10^8$ per second. It is not surprising to now understand that the radius of the pore in CLC channels and transporters is not very different.

The structures of CLC-1 and CLC-K channels support the idea that CLC channels are 'broken transporters' (*Jayaram et al., 2008*; *Lísal and Maduke, 2008*; *Miller, 2006*), where their channel function is built upon a transporter structure with modifications of the gates. The structures demonstrate that relatively small changes in the active site and ion transport pathway of a transporter gives rise to channel function.

# Materials and methods

## Key resources table

| Reagent type (species) or resource | Designation | Source or reference | Identifiers | Additional information |
|---|---|---|---|---|
| Gene (*Homo sapiens*) | CLCN1 | Synthetic | UniProt: P35523 | |
| Cell line (*Homo sapiens*) | HEK293S GnTI⁻ | ATCC | ATCC: CRL-3022 RRID:CVCL_A785 | |
| Cell line (*Spodoptera frugiperda*) | Sf9 | ATCC | ATCC: CRL-1711 RRID:CVCL_0549 | |
| Recombinant DNA reagent | pEG BacMam | doi: 10.1038/nprot.2014.173 | | |
| Software, algorithm | RELION-2 | doi: 10.1016/j.jsb.2012.09.006 | | https://www2.mrc-lmb.cam.ac.uk/relion/index.php?title=Main_Page |
| Software, algorithm | MotionCor2 | doi:10.1038/nmeth.4193 | | http://msg.ucsf.edu/em/software/motioncor2.html |
| Software, algorithm | CTFFIND4 | 10.1016/j.jsb.2015.08.008 | | http://grigorefflab.janelia.org/ctffind4 |
| Software, algorithm | Alightpart_lmbfgs | doi: 10.1016/j.jsb.2015.08.007 | | https://sites.google.com/site/rubinsteingroup/direct-detector-align_lmbfgs |
| Software, algorithm | Rosetta | RosettaCommons | RRID:SCR_015701 | https://www.rosettacommons.org/software |
| Software, algorithm | Pymol | PyMOL Molecular Graphics System, Schrödinger, LLC | RRID:SCR_000305 | http://www.pymol.org/ |
| Software, algorithm | UCSF Chimera | UCSF Resource for Biocomputing, Visualization, and Bioinformatics | RRID:SCR_004097 | http://plato.cgl.ucsf.edu/chimera/ |

## Protein expression and purification

Human CLC-1 was expressed in HEK293 GnTI⁻ cells (ATCC CRL-3022) by transduction using a modified baculovirus as described previously (*Goehring et al., 2014*; *Park et al., 2017*). A human CLC-1 coding sequence (CDS) was synthesized and inserted into a modified pFastBac vector, which contains a CMV promoter upstream of CDS. The expressed CLC-1 construct has a truncation of N-terminal 80 amino acids (residues 2–80), which were predicted to be unstructured, and its C-terminus is fused to enhanced green fluorescent protein (eGFP) (it also contains a HRV 3C protease cleavage sequence between CLC-1 and eGFP). The vector was used for transformation of DH10Bac *E. coli*

cells (Invitrogen) to generate a baculovirus bacmid. Baculoviruses were produced by transfecting *Spodoptera frugiperda* (Sf9; ATCC CRL-1711) cells with the bacmid using Cellfectin-II (Invitrogen). Viruses were then amplified twice for large-scale transduction. HEK293 GnTI⁻ cells were grown at 37°C in suspension in Freestyle 293 medium (Invitrogen) supplemented 2% FBS in the presence of 8% $CO_2$. At a cell density of ~$2.5 \times 10^6$ mL$^{-1}$, baculovirus was added to the culture (6–8% v/v). After incubating at 37°C for ~0.5 day, the culture was supplemented with 10 mM sodium butyrate, then further incubated at 30°C for 2 days before harvest.

All protein purification steps were carried out at 4°C. Harvested HEK293 cells (typically from 1 to 2 L) were suspended in a buffer containing 50 mM Tris-HCl pH 7.5, 300 mM NaCl, 1 mM dithiothreitol (DTT), 1 mM ethylenediaminetetraacetic acid (EDTA), and 10% v/v glycerol, and supplemented with protease inhibitors (50 µM leupeptin, 1 ug/mL aprotinin, 1 uM pepstatin and 1 mM phenylmethylsulfonyl fluoride). 1% dodecyl-β-maltoside (DDM) and 0.2% cholesteryl semisuccinate (CHS) were added to the cell suspension. After extraction for 1.5 h, the lysate was clarified by centrifugation (Beckman Type 70Ti rotor, 40,000 RPM, 1.5 h). The clarified lysate was then mixed with 5 mL of CNBr-sepharose beads (GE Healthcare) coupled with anti-GFP nanobody for 2.5 h. Beads were washed on 60 mL of the buffer containing 0.04% DDM and 0.004% CHS. Bound protein was released from beads by overnight incubation with 5 mL buffer containing 0.04% DDM, 0.004% CHS, and 0.2 mg HRV 3C protease. The retrieved protein was concentrated to 0.5–1.0 mL using Amicon Ultra (100 kDa cutoff; EMD Millipore) and applied to a Superose 6 300/10 GL column (GE Healthcare) equilibrated with 20 mM Tris-HCl pH 7.5, 100 mM NaCl, 1 mM DTT, 0.5 mM EDTA, 0.04% DDM, and 0.004% CHS. The peak fractions were pooled and concentrated to ~4 mg/mL, and immediately used for cryo-EM grid preparation.

The wild-type and mutant *E. coli* CLC transporter proteins (EcCLC) were expressed and purified essentially as described previously (*Dutzler et al., 2002*). *E. coli* BL21 (DE3) (Novagen) was transformed by pET28b vector containing EcCLC CDS, the C-terminal of which was fused to a hexa-histidine tag (His-tag). *E. coli* cells were grown at 37°C in Luria broth (LB) medium containing 60 µg/mL kanamycin until they reached OD600 of 1.2. The expression was induced by addition of 0.2 mM isopropyl β-D-1-thiogalactopyranoside (IPTG). The cells were further grown at 21°C for ~16 h before harvest by centrifugation. The cell pellets were frozen with liquid $N_2$ and stored at −80°C until purification. The frozen *E. coli* cells (typically from 3 L) were thawed and suspended in 20 mM Tris-HCl pH 7.5 and 150 mM NaCl. The cells were lysed by sonication (1 mM PMSF and 50 uM leupeptin were supplemented before the lysis), and 2% decylmaltoside (DM) and 10 mM imidazole were added. After 2-h gentle stirring at 4°C, the lysate was spun for 1 h at 15,000 rpm (Beckman JA-17 rotor). The supernatant was mixed with 5 mL of Talon cobalt agarose beads (Takara Bio) for 2 h. The beads were packed in a column and washed with 25 mL of lysis buffer containing 20 mM imidazole, 12.5 mL of buffer containing 30 mM imidazole, and then 12.5 mL of buffer containing 40 mM imidazole. The protein was eluted by buffer containing 200 mM imidazole. The eluate was concentrated to ~0.5 mL using Amicon Ultra (50 kDa cutoff). The His-tag was removed by adding 0.5 U of Lys-C endopeptidase (Roche) and incubating the mixture at 23°C for 3 h. The eluate was applied to a Superdex 200 300/10 GL column (GE Healthcare) equilibrated with 25 mM Tris-HCl pH 7.5, 100 mM NaCl, 1 mM DTT, 0.5 mM EDTA, 10% glycerol, and 0.3% DM. The peak fraction was collected and used for reconstitution without freezing.

## Cryo-EM analysis

3 µL of purified CLC-1 protein was applied to a glow-discharged gold (or copper for the third dataset) Quantifoil R 1.2/1.3 holey carbon grids (Quantifoil) and incubated for 15 s. Grids were then blotted for 1.5–2.0 s at 4°C and 90% humidity and plunge-frozen in liquid-nitrogen-cooled liquid using Vitrobot Mark III (FEI).

The data sets were collected on a Titan Krios electron microscope (FEI) operated at an acceleration voltage of 300 kV. Dose-fractionated images were recorded on a K2 Summit direct electron detector (Gatan) operated in super-resolution counting mode (a super-resolution pixel size of 0.515 Å) using SerialEM software (*Mastronarde, 2005*). For the first two datasets (2293 movies), the dose rate was 8 e⁻ per pixel per s, and total exposure time was 10 s with 0.2 s for each frame (total cumulative dose of ~75 e⁻ per Å$^2$ over 50 frames). For the third dataset (1998 movies), the dose rate was 5.33 e⁻ per pixel per s, and total exposure time was 15 s with 0.15 s for each frame (total cumulative dose of ~75 e⁻ per Å$^2$ over 100 frames). Defocus values were set from −0.8 µm to −2.4 µm.

Dose-fractionated movies were corrected for gain and motion by MotionCor2 (*Zheng et al., 2017*). Also the pixels were binned to 1.03 Å/pixel during this process. Defocus values were estimated using CTFFIND4 (*Rohou and Grigorieff, 2015*) on the summed micrographs produced by MotionCor2 (using the full dose). Particles were picked automatically by RELION2 (*Kimanius et al., 2016*; *Scheres, 2012*), and obvious artifacts, such as ice contamination and carbon foil, were removed by manual inspection. Total 725,959 particles were extracted with a box size of 320 pixels and subjected to reference-free 2D classification (performed separately per dataset). Based on visual inspection of quality of 2D average classes, 411,260 particles were pooled. This particle set was then applied to the alignpart_lmbfgs program (*Rubinstein and Brubaker, 2015*) to perform per-particle motion correction (particle polishing). The particle polishing step was done using motion-corrected (whole-frame-only) movie stacks, which were first produced by MotionCor2 and then 2x or 4x frame-binned by relion_image_handler (resulting in a total of 25 frames per movie and 3 e$^-$ per Å$^2$ per frame). Particles were extracted from 1 to 13 frames (total dose of 39 e$^-$ per Å$^2$) and using align-parts_lmbfgs's exposure filter. The 'polished' particles were subjected to another round of clean-up by RELION 2D classification (resulting in 350,750 particles). The initial model was generated by RELION auto-refine using particle images from the first dataset and a 50 Å lowpass-filtered model from the CLC-K channel density map (excluding antibody fragments; (*Park et al., 2017*)). All 350,750 polished particle images were subjected to auto-refine (RELION 2.1), using the updated initial model and a soft mask surrounding the protein and detergent micelle density. This refinement step produced a 3.8 Å map (*Figure 2—figure supplement 1B*). This was then followed by a RELION 3D classification procedure skipping image alignment (sorting into five classes). Particles from two classes were combined (175,613 particles) by visual inspection in UCSF Chimera (*Pettersen et al., 2004*) and subjected to RELION auto-refine again. During the later iterations (upon entering the local search mode), the soft mask was updated to contain only the transmembrane or cytosolic domain (focused refinement). The resolution of the final TMD domain map (3.36 Å) was estimated by RELION based on gold-standard Fourier shell correlation (FSC) of independently refined half maps (using the 0.143 cut-off criterion). The focused refinement of the cytosolic domain was performed by 2 iterations of local refinement using reference maps in which information at lower than 4.6 Å resolution were combined from the previous iteration's two half maps. The nominal resolution of the final CTD map is 4.1 Å, but this is likely somewhat overestimated (the resolution before the focused refinement is 4.5 Å). Local resolution was estimated using RELION2's postprocess program (*Figure 2—figure supplement 1A*). Unless stated otherwise, the TMD map shown in figures is a combined map, which was sharpened (B-factor of −97.9 Å$^2$) and lowpass-filtered at 3.36 Å by RELION's automatic postprocess procedure using user-provided soft masks. The TMD map in *Figure 2C* and *Videos 1* and *2* was sharpened with a B-factor of −97.9 Å$^2$ and low-pass filtered at 3.1 Å. The CTD density map was low-pass filtered at 4.2 Å without B-factor sharpening.

## Atomic model building

An initial model of the CLC-1's TMD was generated by the SWISS-MODEL homology modelling webserver using the CLC-K model (PDB ID: 5TQQ) as a template. The output model was fit into the TMD density map using Chimera and rebuilt using Coot (*Emsley et al., 2010*). Model refinement was done in real space using Rosetta 3.7 using a script developed for cryo-EM model refinements (*Wang et al., 2016*) (*Table 1*). The first round was performed with an asymmetric unit model, and the five best output models were selected based on Rosetta's energy scores. A consensus model was generated by combining fragments from these models based on the fit to the density map. The subsequent two rounds of Rosetta refinement were done with two-fold symmetry imposed. To prevent overfitting, the weight between Rosetta energy scores and the fit to the experimental density map was adjusted, and test refinement was performed on one of two half maps. The output models were then compared to both half maps by calculating FSC (*Figure 2—figure supplement 2*). To this end, we used a weight of 25, which gave us a good fitting to the map and negligible overfitting. While the first two rounds of refinement were done using one of the two half maps, the last round was performed on the combined map to maximize the use of experimental data in refining the model (see *Figure 2—figure supplement 2* for FSC between the final model and the combined map). The final model was selected among ~2000 Rosetta-generated models based on Rosetta's total score (top 20%) and the fit of side chains to the map (visual inspection). No further modifications were made except for Cl$^-$ ions at S$_{ext}$ and S$_{int}$, which were modelled in Coot (Coot's real-space

**Table 1.** Model refinement and validation statistics.

| | TMD | CTD |
|---|---|---|
| Rosetta Model Refinement | | |
| Map pixel size (Å) | 1.03 | 1.03 |
| Map sharpening B-factor (Å$^2$) | −97.9 | 0 |
| Map lowpass filter (Å) | 3.36 | 4.2 |
| Refinement resolution limit (Å) | 3.36 | 4.5 |
| Number of atoms | 14,536 | 5124[†] |
| Protein | 14,536 | 5124[†] |
| Non-hydrogen atoms | 7152 | 2550[†] |
| Hydrogen atoms | 7384 | 2574[†] |
| Non-protein | 0 | 0 |
| **Refined Model Statistics** | | |
| Average B-factor (Å$^2$) | 24.59 | 161.34 |
| r.m.s deviations | | |
| Bond length (Å) | 0.02 | 0.02[†] |
| Bond angle (°) | 1.42 | 1.55[†] |
| Ramachandran Plot | | |
| Favored (%) | 96.75 | 96.15[†] |
| Outliers (%) | 0.43 | 0.64[†] |
| MolProbity | | |
| Clash score*/percentile | 1.38 (99 %) | 0.39[#] (99%) |
| Rotamers | | |
| Favored (%) | 99.48% | 100.00[#] % |
| Outliers (%) | 0.00% | 0.0[†] % |
| Overall score/percentile | 1.07 (100 %) | 0.90[†] (100%) |

*number of steric overlaps >0.4 Å per 1000 atoms.

[†]numbers and scores before truncation of side chain atoms.

DOI: https://doi.org/10.7554/eLife.36629.019

refinement was used) since Rosetta could not refine Cl$^-$ ions. Modelling of CTD was done similarly using Rosetta, but using a crystal structure of CLC-0 CBS domain as an initial model. A weight of 7 was used, and the refinement was limited to 4.5 Å resolution. As side chains were not visible in the CTD density map, we removed all side chain atoms from the final CTD model generated by Rosetta. The following segments were not modelled as they were invisible in the density maps: N–115, 251–262 (a cytosolic segment between αF and αG), 671–796 (a loop in CTD), and 877–988(C). MolProbity was used for structural validation of models (*Table 1*) (*Chen et al., 2010*).

Detection of pores and estimation of pore radii (*Figures 3* and *6*) were performed using Caver (*Chovancova et al., 2012*). In the case of EcCLC (PDB ID: 1OTS) and CmCLC (PDB ID: 3ORG), Glu$_{gate}$ (E148 of EcCLC and E210 of CmCLC) was mutated to Ala before estimation since its side chain is blocking the Cl$^-$ pathway. In the case of bovine CLC-K (PDB ID: 5TTQ) (*Figure 6B*), we changed the rotamer conformation of V166 (equivalent to Glu$_{gate}$) from original *gauche+* (63°) to *trans* (175°). With the original rotamer, the constriction around S$_{ext}$ was found to be too narrow (radius < 0.9 Å) for pore detection. Because both rotamers can be fitted equally well into the cryo-EM density map, it is uncertain which is right or if both can exist in the protein. We note that *trans* is in general an energetically more favored rotamer than *gauche+*. Water accessibility in CLC-1's vestibules (*Figure 4B*) was probed using HOLLOW (*Ho and Gruswitz, 2008*) using a probe radius of 1.4 Å. Protein electrostatics were calculated using the Adaptive Poisson-Boltzmann Solver (*Baker et al., 2001*) with a parameter of 150 mM monovalent salt concentration. UCSF Chimera and PyMOL (Schrödinger) were used to prepare structure figures.

## Reconstitution of E. coli CLC transporter mutants and flux assays

To reconstitute EcCLC mutant proteins for $Cl^-$ efflux assays, *E. coli* polar lipids in chloroform (Avanti Polar Lipids) was dried in a glass tube with an argon stream, followed by overnight incubation in a vacuum chamber. Dried lipids were suspended by sonication in buffer (RB-Cit) containing 25 mM sodium citrate (pH 4.6) and 300 mM KCl and then solubilized with 35 mM (3-((3-cholamidopropyl) dimethylammonio)−1-propanesulfonate) (CHAPS; Anatrace) and additional sonication. Purified EcCLC protein was added to the lipid/CHAPS mixture in a protein-to-lipid ratio of 1:5000 (wt:wt). After 30 min incubation at 23°C, the mixture was dialyzed against RB-Cit buffer to remove CHAPS. The dialysis was carried out at 4–10°C for 48 h with three additional buffer changes. To reconstitute EcCLC proteins for fluorescence-based flux assays, the same procedure was used except that buffer containing 10 mM HEPES-NaOH (pH 7.0) and 450 mM KCl instead of RB-Cit and a protein-to-lipid ratio of 1:500 (wt:wt) were used. After dialysis, proteoliposome vesicles were aliquoted, flash-frozen with liquid $N_2$, and stored at −80°C until use.

The $Cl^-$ efflux (dump) assays were performed essentially as described previously (*Jayaram et al., 2008*; *Walden et al., 2007*). A frozen aliquot of vesicles was thawed and briefly sonicated in the bath sonicator (Branson). Vesicles were extruded through a 0.4 µm polycarbonate filter 19 times (Avanti Mini-Extruder). The extruded vesicles were desalted with a spin column packed with Sephadex G-50 resin (~2.5 mL bed volume) equilibrated with buffer (EB) containing 25 mM sodium citrate (pH 4.7), 250 mM $K_2SO_4$, and 1 mM NaCl. 100 µL of the desalted vesicles were then mixed with 900 µL EB in a chamber equipped with a magnetic stirrer and $Cl^-$-selective electrode (Fisher Accumet). Changes in extravesicular $Cl^-$ concentration was monitored over time by the $Cl^-$-selective electrode connected to a computer through a digitizer (DataQ). To calibrate the electrode, 0.1 mM NaCl was added before the vesicles were added. Flux was initiated by addition of 2 µg/mL valinomycin and 1 µg/mL carbonyl cyanide-p-trifluoromethoxyphenylhydrazone (FCCP) or 3 µg/mL valinomycin (for E148A mutants). At the end of assays, 30 mM octyl β-glucoside (Anatrace) was added to release all $Cl^-$ content from vesicles. Calculation of $Cl^-$ transport rates were carried out as described previously (*Walden et al., 2007*). Volume changes by dialysis, extrusion, and desalting steps were included in calculation.

The fluorescence-based flux assays were performed as follows based on (*Feng et al., 2012*). A frozen aliquot of vesicles was thawed and briefly sonicated in a bath sonicator. 3 µL of vesicles were mixed with 40 µL of assay buffer containing 20 mM HEPES-NaOH (pH 7.0), 450 mM K-gluconate and 4 µM ACMA in a well of a 384-well fluorescence assay. After measuring initial AMCA fluorescence intensity ($\lambda_{Ex}$=410 nm, $\lambda_{Em}$=490 nm), $Cl^-/H^+$ flux was initiated by addition of 1 µM valinomycin, followed by monitoring fluorescence over time (10 s intervals) using a plate reader (Tecan Infinite M1000) at 27°C. Note that there is a dead time for measurement between t = 80 s to t = 120 s due to handling of the plate during valinomycin addition. Valinomycin was added to the reactions at t = ~100 s. As a control, 0.9 µM carbonyl cyanide 3-chlorophenylhydrazone (CCCP) was added to the assay mixture at the end of the experiment to dissipate an accumulated $H^+$ gradient. To measure relative $H^+$ transport activity of each mutant, time required to fluorescence reaches 75% (or 85% in the case of S106G/F348A double mutant) of the initial fluorescence upon addition of valinomycin was calculated. This time value was then inversed and normalized with respect to a value obtained with wild-type EcCLC.

## Data availability

Cryo-EM density maps of human CLC-1 have been deposited in the electron microscopy data bank under accession code EMD-7544 and 7545. Atomic coordinates have been deposited in the protein data bank under accession code 6COY and 6COZ.

## Acknowledgements

We thank M Ebrahim and J Sotiris at the Rockefeller University Cryo-EM resource center for help with microscope operation, YC Hsiung for help with large-scale cell culture, members of the MacKinnon lab for helpful discussions. EP was supported by the Jane Coffin Childs Memorial Fund fellowship (#61–1513) and Charles H Revson fellowship (#16–33). RM is a Howard Hughes Medical Institute investigator.

## Additional information

### Funding

| Funder | Grant reference number | Author |
|---|---|---|
| Howard Hughes Medical Institute | | Roderick MacKinnon |
| Jane Coffin Childs Memorial Fund for Medical Research | 61-1513 | Eunyong Park |
| Charles H. Revson Foundation | 16-33 | Eunyong Park |

The funders had no role in study design, data collection and interpretation, or the decision to submit the work for publication.

### Author contributions

Eunyong Park, Conceptualization, Formal analysis, Funding acquisition, Investigation, Visualization, Writing—original draft, Writing—review and editing; Roderick MacKinnon, Conceptualization, Formal analysis, Supervision, Funding acquisition, Writing—original draft, Writing—review and editing

### Author ORCIDs

Eunyong Park (iD) https://orcid.org/0000-0003-2994-5174
Roderick MacKinnon (iD) http://orcid.org/0000-0001-7605-4679

### Decision letter and Author response

Decision letter https://doi.org/10.7554/eLife.36629.030
Author response https://doi.org/10.7554/eLife.36629.031

## Additional files

### Supplementary files

• Transparent reporting form
DOI: https://doi.org/10.7554/eLife.36629.020

### Data availability

Cryo-EM density maps of human CLC-1 have been deposited in the electron microscopy data bank under accession code EMD-7544 and 7545. Atomic coordinates have been deposited in the protein data bank under accession code 6COY and 6COZ.

The following datasets were generated:

| Author(s) | Year | Dataset title | Dataset URL | Database, license, and accessibility information |
|---|---|---|---|---|
| Park E, MacKinnon R | 2018 | Human CLC-1 chloride ion channel, transmembrane domain | https://www.rcsb.org/structure/6COY | Publicly available at the RCSB Protein Data Bank (accession no. 6COY) |
| Park E, MacKinnon R | 2018 | Human CLC-1 chloride ion channel, C-terminal cytosolic domain | https://www.rcsb.org/structure/6COZ | Publicly available at the RCSB Protein Data Bank (accession no. 6COZ) |
| Park E, MacKinnon R | 2018 | Human CLC-1 chloride ion channel, transmembrane domain | http://www.ebi.ac.uk/pdbe/entry/emdb/EMD-7544 | Publicly available at the Electron Microscopy Data Bank (accession no. EMD-7544) |
| Park E, MacKinnon R | 2018 | Human CLC-1 chloride ion channel, C-terminal cytosolic domain | http://www.ebi.ac.uk/pdbe/entry/emdb/EMD-7545 | Publicly available at the Electron Microscopy Data Bank (accession no. |

EMD-7545)

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
