## [Decision Letter]

Thank you for submitting your article "Structure of the CLC-1 chloride channel from Homo sapiens" for consideration by *eLife*. Your article has been favorably evaluated by Richard Aldrich (Senior Editor) and three reviewers, one of whom, László Csanády (Reviewer #1), is a member of our Board of Reviewing Editors. The following individuals involved in review of your submission have agreed to reveal their identity: Alessio Accardi (Reviewer #2); Christopher Miller (Reviewer #3).

The reviewers have discussed the reviews with one another and the Reviewing Editor has drafted this decision to help you prepare a revised submission.

Summary:

This manuscript by Park and Mackinnon presents the first high-resolution (~3.4Å) structure of a human CLC channel, hCLC-1, solved by electron-cryomicroscopy. The overall architecture of the homodimeric protein is very similar to those of other solved CLC-family structures, and includes in each subunit a pore which is bifurcated on the cytosolic side and harbors three anion binding sites (S_ext_, S_cen_, S_int_). On the other hand, subtle differences in pore diameter and the side-chain chemistry of pore-lining residues helps to understand the very different functional properties of members of the channel and transporter subclass of the CLC family. The authors propose that channel-like behavior is due to (i) a reduced kinetic barrier for Cl^-^ flux between S_cen_ and S_int_, (ii) low-affinity anion binding in S_cen_, and (iii) a permissive (off-pore) position of the side-chain of the "gating glutamate" (Glu_gate_). These predictions are elegantly supported by functional analysis of a bacterial CLC transporter (EcCLC) mutated at various key positions. Overall, this is an excellent manuscript with no major concerns noted by the reviewers. Below are suggestions for three points that would benefit from a few more sentences of discussion, to help harmonize their mechanistic model with pre-existing data and literature.

Essential revisions:

1) Despite the off-pore position of Glu_gate_, the diameter of the external pore entrance is still too narrow to accommodate an ion. While this is not shocking, as all the CLC structures share this feature, it is still puzzling how such a pore can conduct ions at a rate of ~10^6^ ions s^-1^. Along these lines, it would be helpful if the authors could discuss a bit more on Cl^-^ throughput rates, and say something about the issue of electro-diffusive vs. uniporter-like Cl^-^ flux.

2) CLC-1 is activated by extracellular pH. The current consensus open pore was Glu_gate_ in the 'up' (E148Q-like structure of CLC-ec1) conformation. This explained well how CLC-1 could be activated by e.c. low pH, as Glu_gate_ would be accessible to water and H^+^ from the extracellular solution and stabilized in this state. If the sideways conformation of the Glu_gate_ side chain seen in the new structure indeed reflects the open pore, then it is hard to envision how that side chain is protonated from the extracellular side, and how protonation would stabilize this state. Thus, it would be helpful if the authors could comment on the mechanism of CLC-1 activation by extracellular low pH.

3) The cartoon in Figure 7C suggests that the Glu_gate_ side chain is itself the fast protopore gate. However, E-to-A mutations of Glu_gate_ in Clc-0 (Dutzler et al., 2003), and evolutionary removal of Glu_gate_ in Clc-K (L'Hoste et al., 2013, PMID: 23791703), result in channels in which the protopores are still seen to gate (open and close), albeit with high open probabilities. Wouldn't this suggest that the fast gate is not simply formed by the Glu_gate_ side chain? Could the brief fluctuations of those channels reflect small movements of the pore lining residues that temporarily block permeation through the narrow channel pore?

---

## [Author Response]

Essential revisions:1) Despite the off-pore position of Glu_gate_, the diameter of the external pore entrance is still too narrow to accommodate an ion. While this is not shocking, as all the CLC structures share this feature, it is still puzzling how such a pore can conduct ions at a rate of ~10^6^ ions s^-1^. Along these lines, it would be helpful if the authors could discuss a bit more on Cl^-^ throughput rates, and say something about the issue of electro-diffusive vs. uniporter-like Cl^-^ flux.

We added an additional paragraph in the Discussion (sixth paragraph) to address this issue. In brief, we think scientists should change their expectations regarding pore diameter and throughput. Now with data to compare two CLC channel structures and multiple CLC transporter structures we can place data ahead of preconceived notion. We come to the same conclusion independently through the study of other ion channels. Take for example K^+^ channels, in which the selectivity filter between ion binding sites is narrower than the diameter of a K^+^ ion. In the added paragraph we explain that the chemical and electrostatic properties of atoms lining the pore and their dynamic properties (ability to move out of the way) will affect conductivity. This is in fact an old problem in biochemistry considered in other contexts such as diffusion of oxygen to the heme co-factor inside a protein.

2) CLC-1 is activated by extracellular pH. The current consensus open pore was Glu_gate_ in the 'up' (E148Q-like structure of CLC-ec1) conformation. This explained well how CLC-1 could be activated by e.c. low pH, as Glu_gate_ would be accessible to water and H^+^ from the extracellular solution and stabilized in this state. If the sideways conformation of the Glu_gate_ side chain seen in the new structure indeed reflects the open pore, then it is hard to envision how that side chain is protonated from the extracellular side, and how protonation would stabilize this state. Thus, it would be helpful if the authors could comment on the mechanism of CLC-1 activation by extracellular low pH.

We added a paragraph (Discussion, third paragraph) discussing this issue. As described there are several possible explanations but the bottom line is we do not understand this yet.

3) The cartoon in Figure 7C suggests that the Glu_gate_ side chain is itself the fast protopore gate. However, E-to-A mutations of Glu_gate_ in Clc-0 (Dutzler et al., 2003), and evolutionary removal of Glu_gate_ in Clc-K (L'Hoste et al., 2013, PMID: 23791703), result in channels in which the protopores are still seen to gate (open and close), albeit with high open probabilities. Wouldn't this suggest that the fast gate is not simply formed by the Glu_gate_ side chain? Could the brief fluctuations of those channels reflect small movements of the pore lining residues that temporarily block permeation through the narrow channel pore?

The evidence that Glu_gate_ dominates protopore gating is strongly supported by data. When Glu_gate_ is mutated you do see some residual (much less frequent) gating/blocking events that are likely due to other changes in pore conformation or block (likely, because almost all ion channels exhibit such closing events). Therefore, no, these observations do not suggest to us that the fast gate is not formed by Glu_gate_.